



# Concurrent Satellite and ground-based Lightning Observations from the Optical Lightning Imaging Sensor (ISS-LIS), the LF network Meteorage and the SAETTA LMA in the northwestern Mediterranean region

Felix Erdmann[1,2], Eric Defer[1], Olivier Caumont[2], Richard J. Blakeslee[3], Stéphane Pédeboy[4], and Sylvain Coquillat[1]

[1]Laboratoire d'Aérologie, Université de Toulouse, CNRS, UPS, Toulouse, France
[2]CNRM, Université de Toulouse, Météo-France, CNRS, Toulouse, France
[3]NASA George C. Marshall Space Flight Center/NSSTC, Huntsville, AL, USA
[4]Météorage, Pau, France

**Correspondence:** Felix Erdmann (erdmann.professional@gmx.de)

**Abstract.** The new space-based Lightning Imager (LI) on board the Meteosat Third Generation (MTG) geostationary satellite will improve the observation of lightning over Europe, the Mediterranean Sea, Africa and the Atlantic Ocean from 2021 onwards. In preparation of the use of the upcoming MTG-LI data, we compare observations by the Lightning Imaging Sensor (LIS) on the International Space Station (ISS), which applies an optical technique similar to MTG-LI, to concurrent records of

the Low Frequency (LF) ground-based network Meteorage. Data were analyzed over the northwestern Mediterranean Sea from March 01, 2017 to March 20, 2018. Flashes are detected by ISS-LIS using illuminated pixels, also called events, within a given (2.0 ms) frame and during successive frames. Meteorage describes flashes as a suite of Intra-Cloud/cloud-to-cloud (IC) pulses and/or Cloud-to-Ground (CG) strokes. Both events and pulses/strokes are grouped to flashes using a novel in-house algorithm.

In our study, ISS-LIS detects about 57 % of the flashes detected by Meteorage. The flash detection efficiency (DE) of
Meteorage relative to ISS-LIS exceeds 80 %. Coincident matched flashes detected by the two instruments show a good spatial and temporal agreement. Both peak and mean distance between matches are smaller than the ISS-LIS pixel resolution (about 5.0 km). The timing offset for matched ISS-LIS and Meteorage flashes is usually shorter than the ISS-LIS integration time frame (2.0 ms). The closest events and pulses/strokes of matched flashes achieve sub-millisecond offsets. Further analysis of flash characteristics reveals that longer lasting and more spatially extended flashes are more likely detected by both ISS-LIS
and Meteorage than shorter duration and smaller extent flashes. ISS-LIS' relative DE is lower for daytime versus nighttime as well as for CG versus IC flashes.

A second ground-based network, the Very High Frequency (VHF) SAETTA Lightning Mapping Array (LMA), further enhances and validates the lightning pairing between ISS-LIS and Meteorage. It also provides altitude information of the lightning discharges and adds a detailed lightning mapping to the comparison for verification and better understanding of the
processes. Both ISS-LIS and Meteorage flash detections feature a high degree of correlation with the SAETTA observations (without altitude information). In addition, Meteorage flashes with ISS-LIS match tend to be associated with discharges that





occur at significantly higher altitudes than unmatched flashes. Hence, ISS-LIS flash detection suffers degradation with low-level flashes resulting in lower DE.

# 1 Introduction

Lightning defines electrical discharges within the atmosphere. The discharges can happen within a cloud or between clouds (IC) or between a cloud and the ground (CG). The total lightning activity is of interest for e.g. the numerical weather prediction (NWP) as lightning serves as tracer for deep convection. Among others Mattos et al. (2017) investigated the life cycle of thunderstorms and processes leading to the different discharge types. They found in their analysis of 46 isolated thunderstorms that in 98% of their cases, the first CG flash is preceded by IC lightning by approximately six minutes on average. To maximize

the impact of lightning data on the assimilation in NWP systems, total lightning should be observed continuously over large areas.

At this time, lightning observations in Europe use mainly ground-based sensors. In a few years, the new Lightning Imager (LI) on board the Meteosat Third Generation (MTG) satellite (Stuhlmann et al., 2005) will provide continuous lightning observation over Europe, the Mediterranean Sea, Africa, the Atlantic Ocean and parts of Brazil. The satellite sensor will be

able to detect the total lightning including CG and IC flashes when it is launched in the 2021 time frame. The Lightning Imaging Sensor (LIS) on the International Space Station (ISS) (Blakeslee and Koshak, 2016) creates a unique opportunity to provide proxy data to help prepare research and operational applications for the MTG-LI data. It overpasses, among others, wide parts of Europe, including the entire Mediterranean region. ISS-LIS is in principle similar to the planned MTG-LI, so that ISS-LIS data can to some extent mimic the upcoming MTG-LI data. In addition, a comparison between European ground-

based lightning observation networks and ISS-LIS should improve the understanding of ground- and space-based lightning observation. All instruments and networks are hereafter referred to as lightning locating systems (LLSs) bearing in mind ISS-LIS and SAETTA do actually map lightning.

A LIS instrument was previously operational on the Tropical Rainfall Measurement Mission (TRMM) satellite (e.g., Christian et al., 1999; Cecil et al., 2005). Several LLS comparisons exist for regions covered by TRMM-LIS. The focus of the

following (not exhaustive) literature review is on observational analyses rather than laboratory experiments, e.g. Boccippio et al. (2002). There are a variety of ground-based LLSs that have been utilized:

Very low frequency (VLF) and low frequency (LF) LLSs can detect lightning over hundreds or thousands of kilometers with the drawback of mainly detecting CG return strokes and hence a limited overall detection efficiency (DE).

Aiming at exploring suitable proxy data for the Geostationary Lightning Mapper (GLM) (Goodman et al., 2013), Thompson

et al. (2014) report a pulse/stroke DE maximum for two long range LLSs, the World Wide Lightning Location Network (WWLLN) and the Earth Networks Total Lightning Location Network (ENTLN), of 18.9 % and 63.3 %, respectively, relative to 18-months records of TRMM-LIS groups. The maxima were found over the Pacific Ocean for WWLLN and near North America for ENTLN (within the analyzed region with the highest sensor density) in 2010 and 2011. They did not study how many WWLLN and ENTLN pulses/strokes had coincident TRMM-LIS groups.





Rudlosky et al. (2017) analyzed the performance of the Global Lightning Dataset 360 (GLD360) relative to TRMM-LIS from 2012 to 2014 in different regions. GLD360 was able to detect 63.6 % of the TRMM-LIS flashes in North America in 2014, the maximum DE reported in their study. The performance steadily increased from 2012 to 2014. The relative DE of GLD360 increased with the TRMM-LIS flash duration, flash extent and group number. The mean (median) location offset of the nearest GLD360 stroke to the matched TRMM-LIS flashes was 8.7 km (7.0 km). Rudlosky et al. (2017) applied the assumption that TRMM-LIS would detect all flashes within its field of view, but did not study the reverse problem, i.e. the relative DE of TRMM-LIS to the GLD360 flashes or strokes.

Defer et al. (2005) used both the UK Met Office long-range VLF sferics ATD system and TRMM-LIS to study the lightning activity in the eastern Mediterranean Sea for 20 days during winter 2008-2009. For their investigation of the flash scale, they developed and employed their own algorithm for TRMM-LIS flashes. The flash density analysis exhibits a general agreement between ATD and TRMM-LIS. The relatively small data set, and the fact that ATD detected mostly CG lightning, limited the ability for gaining overall statistics.

Bitzer et al. (2016) tested a Bayesian approach on the DE of TRMM-LIS and ENTLN by implementing the conditional DEs of the two LLSs relative to each other. They found a relative conditional group-to-pulse DE of 52 % (27 %) for TRMM-LIS to ENTLN (ENTLN to TRMM-LIS near North America in 2013). They also addressed peak timing differences and distances for the collocated discharges (again LIS groups, ENTLN pulses; see section 3.2 for details). Bitzer et al. tested further the effect of assimilating one dataset into the other on the detected number of discharges, i.e. 23.6 % of discharges could be added to TRMM-LIS records.

While the previous papers focused on the DE, Höller and Betz (2010) analyzed TRMM-LIS and a VLF/LF lightning location network (LINET) in order to generate random proxy optical data from a given set of LINET data using model distribution functions. The outcomes are of specific interest for proxy data for the MTG-LI. Besides the relative DEs (approximately 50 % for both LLSs), they investigated distribution functions and correlations between TRMM-LIS group and LINET pulse/stroke number per flash, flash extent and duration and between LINET pulse/stroke amplitude and TRMM-LIS group radiance. Although the Pearson correlation coefficients remained low, the approach can be further refined for high fidelity MTG-LI proxy data.

A second type of ground-based LLSs uses very high frequency (VHF). Recorded signals and related physical processes can differ from those recorded by VLF/LF LLSs. VHF LLSs typically feature high DE performances and three-dimensional (3D) lightning channel mapping (Thomas et al., 2004). Their drawback is the limited range. These LLSs use direct line of sight signal detection, and thus, the range suffers from the Earth's curvature and terrain shading effects. LF LLSs can benefit from a reflection of the LF signal at the ionosphere.

Thomas et al. (2000) presented a case study of a storm in Oklahoma, USA, at local nighttime. The storm was observed by both the local Lightning Mapping Array (LMA) and TRMM-LIS. 108 of the 128 LMA lightning discharges were detected by TRMM-LIS and the LMA detected all TRMM-LIS flashes. The lightning missed by TRMM-LIS was mainly confined to low altitude discharges, i.e. below 7.0 km. Optical signals of lightning discharges that propagated via scattering to the upper part of the cloud were easily detected by TRMM-LIS.



Blakeslee et al. (2002) studied the São Paulo LMA (SP-LMA) dataset and its capability to serve for GLM proxy data. TRMM-LIS events were in good agreement with the concurrent SP-LMA, ENTLN and LINET observations regarding latitude, longitude and timing. The records showed as expected more VHF (SP-LMA) sources than VLF pulses/strokes (ENTLN and LINET) per flash.

Due to TRMM satellite orbits, the comparisons of TRMM-LIS and ground-based LLSs records are restricted to tropical
and subtropical regions between about 38 °N and 38 °S. As a result of its higher inclination orbit, ISS-LIS now allows the observation of extra-tropical thunderstorms to extend to 55 °N and S. The higher latitude storms might show different behaviors to their tropical and subtropical counterparts due to modified cloud vertical extent and forcing like the general wind field, average temperature and temperature gradients. Our study concentrates on the characteristics of lightning flashes over the northwestern (NW) Mediterranean Sea and should contribute to a better understanding of both European storms and European
LLSs. This allows for the first time an intercomparison of LIS and European LLSs. Three LLSs operating in different spectral regions (Near-IR, VLF/LF, VHF) are compared: The satellite-based ISS-LIS being operational since March 2017, the French Meteorage VLF/LF LLS and the VHF SAETTA LMA on Corsica. The relative DE of ISS-LIS to Meteorage (and reverse) is analyzed, while SAETTA is used to verify and understand the results. Indeed, the spatially and temporally high resolution of SAETTA's measurements capture the structure and the life cycle of each lightning flash and gather additional information,
i.e. discharge altitude, to assess more thoroughly ISS-LIS and Meteorage strengths and weaknesses. Among the commonly investigated relative DEs, distances, timing offsets and specific characteristics of matched ISS-LIS and Meteorage flashes are further examined. This work aims at providing the basis for mimicking optical, satellite-based lightning data from a VLF/LF LLS.

In section 2 ISS-LIS, Meteorage and SAETTA are introduced as well as the data processing, developed algorithms and the
investigation methodology. Results are presented in section 3. A brief summary and some discussion are given in section 4.

## 2   Instrumentation and Methodology

This paper aims at identifying the individual characteristics in lightning detection by the satellite-based ISS-LIS, the VLF/LF Meteorage and the VHF SAETTA LLSs. ISS-LIS, installed on the International Space Station in 2017, has been acquiring data since March 01, 2017. Our intercomparison of the LLSs covers the period from March 01, 2017 until March 20, 2018.
The region was limited to 40.5°N to 44.0°N and 7.0°E to 11.0°E around the island of Corsica in the NW Mediterranean Sea. Figure 1 shows the domain with accumulated data of one overpass (a), an infrared (IR) satellite picture (b) and the example of one flash recorded by ISS-LIS, Meteorage and SAETTA (c). The three instruments are introduced within this section. In total, ISS-LIS field of view (FOV) intersected the region of interest 851 times during the study period, with 26 of the overpasses exhibiting lightning activity. In our paper, all times are given in Coordinated Universal Time (UTC). Altitudes are defined
above sea level (ASL). Distances are calculated using Vincenty's formulae (Vincenty, 1975) based on the WGS 84 reference ellipsoid which are more accurate on Earth than for example great circle distances (assumes the Earth as oblate sphere rather than a sphere). The term *detection efficiency* (DE) means in the following the DE for flashes, not event or pulse/stroke DE.






**Figure 1.** Observations of ISS-LIS events (as pixel centers), Meteorage pulses/strokes and SAETTA VHF sources (as indicated) during one ISS overpass over Corsica on Sept. 10, 2017 (a). The ISS-LIS viewtime is presented as grayscale of the background. Numbers in parentheses give the number of SAETTA VHF sources, ISS-LIS events and Meteorage pulses/strokes, respectively. (b) shows the infrared (IR 10.8 µm) satellite image of the same day at 01:15:00 UTC. One flash over Corsica detected by the three LLSs during the same ISS overpass is shown in (c).

## 2.1 ISS-LIS

The ISS operates in Low Earth Orbit (LEO) and overpasses one region on the surface up to three times a day (up to two times in the tropics). Lightning observation of a specific point lasts up to 90 seconds per overpass due to the ISS orbit characteristics and the LIS FOV of approximately 655 x 655 km$^2$. The optical lightning detection is performed at a wavelength of 777.4 nm





at the atomic oxygen line. ISS-LIS observes both IC and CG discharges but cannot distinguish the lightning type. ISS-LIS captures an image of the Earth every two milliseconds referred to as a frame. The LIS focal plane consists of a 128 x 128 pixel Charge Couple Device (CCD) that is read out every 2 ms. The pixel FOV ranges between 4.5 km (nadir) and 6.2 km at the edges

(Dennis Buechler, personal communication 2019). Blakeslee and Koshak (2016) apply a four-step filtering approach, involving spatial, spectral, temporal and background subtraction filter, to identify pixels with lightning activity. This is required to detect the lightning during daytime when the sunlight reflected off the cloud tops otherwise overwhelms and masks the lightning signal (i.e., it is daytime lightning detection that drives the design of space-based lightning detectors such as LIS and the new MTG-LI). An illuminated pixel that breaks a predefined threshold in a given 2 ms frame is identified as an event. Events define

the smallest units of the optical signals in the ISS-LIS data set. Their latitude and longitude correspond to the pixel center. A group is the next unit of ISS-LIS data. An ISS-LIS group contains one or more events occurring within the same time frame and in adjacent pixels of the ISS-LIS image (Christian et al., 2000). Next, groups are organized into flashes, so that a flash can consist of one or multiple groups. Groups occurring within 330 ms and 5.5 km belong to the same flash. The locations of groups and flashes are defined by the radiance weighted average positions of their events and groups, respectively. Finally, an

area contains all flashes with distances of less than 16.5 km to each other. The National Aeronautics and Space Administration (NASA) provides the ISS-LIS in different postprocessing levels. In the latest available version, P0.2, the quality control is already close to its (expected) final stage, but the data may contain some undetected minor errors (Blakeslee et al., 2017). The main difference will concern the detection efficiency. The fully validated flash density should not alter more than 5.0 % to 10.0 % from version P0.2 (R. Blakeslee, personal communication 2018). LIS data comprises the 2 ms scientific data, e.g.

time, latitude, longitude and radiance of events and instrument, platform or external errors to verify the data quality, and housekeeping data. The housekeeping data, received every second, contains among others LIS' viewtime with information about the FOV at a time. It is provided on a 0.5° x 0.5° grid. ISS-LIS viewtime is fundamental for the intercomparison to continious observations at the ground.

The original ISS-LIS data contains times in the International Atomic Time with reference to 01 Jan. 1993 (TAI93) format.

For the intercomparison of the LLSs, times are converted to UTC while taking the missing leap seconds into account. The ISS-LIS times include a time-of-flight (TOF) correction accounting for the time photons need to travel from the lightning discharge on Earth to the satellite.

## 2.2 Meteorage

The Meteorage LF LLS uses Vaisala LS7002 sensors (Vaisala, 2013) at a frequency between 1 kHz and 350 kHz. It includes

21 ground sensors across France and contributes to the European Cooperation for Lightning Detection (EUCLID). EUCLID comprises lightning sensors all over Europe and helps to improve the performance of national LLSs (Schulz et al., 2016). The LS7002 sensors measure the signals related to CG strokes and IC pulses, thus the total lightning. Vaisala claims a CG DE of 95 % and a DE for IC of 50 %. Pédeboy et al. (2018a) stated that indeed 97 % of the CG flashes and 56 % of the IC flashes were detected by Meteorage (68.3 % overall DE relative to LMA flashes). The theoretical median location accuracy approximates

250 m and improves inside the network to about 150 m. Pédeboy et al. (2018a) found reduced median location accuracy for





IC flashes of 1.64 km. Time synchronization applies a GPS receiver with an accuracy of 50 ns to UTC. The lightning location needs at least four sensors by applying combined magnetic direction finding and time-of-arrival techniques. Lightning can be detected in a distance up to 1500 km from a sensor. In practice, the use of ionospheric reflection is avoided, hence, limiting the sensor range to about 625 km. It ensures the ground plane wave front of the signal is measured rather than reflected wave

of the lightning related signal. Our study makes use of the Meteorage lightning pulse/stroke data. For each pulse/stroke, the occurrence time, latitude, longitude, the amplitude with polarity and the type (IC/CG) are provided. Meteorage data are then disregarded if observation space or time do not fit the corresponding ISS-LIS viewtimes.

### 2.3 SAETTA (Suivi de l'Activité Electrique Tridimensionnelle Totale de l'Atmosphère)

The LMA technology was developed by New Mexico Tech (Rison et al., 1999). The SAETTA LMA operates in the 60-

66 MHz VHF band, with an 80 μs analysis window (Coquillat et al., 2014), and consists of 12 LMA stations distributed over the island of Corsica. The distance between the network's northernmost and southernmost (westernmost and easternmost) stations approximates 180 km (70 km). The station altitude ranges from 3.8 mASL to 1950.2 mASL. SAETTA maps the total lightning activity. Fast CG discharges travelling between the cloud and ground in time frames shorter than 80 μs might be missed. A minimum of six stations is needed to capture a lightning source in 3D. Redundant information from more stations

improves the location accuracy and consequently decreases the chance of mislocation and possible noise (e.g. single VHF sources in Figure 1(c)). As a drawback, less VHF sources and flashes are detected simultaneously by more than six stations. Aiming at a high flash DE, coincident signals at six stations are sufficient for the LMA data in this study.

SAETTA data include the time, latitude, longitude, altitude, amplitude of each lightning source. Lightning location reaches up to a radius of 350 km from the center of the network. The SAETTA location uncertainty increases with the distance to

the network center. According to the theoretical model of Thomas et al. (2004), the radial azimuthal and altitude errors are, at best for VHF sources at 10 km altitude, 15 m, 8 m and 40 m, respectively, within 50 km from the center of the network (Coquillat et al., 2019). These theoretical errors reach about 300 m, 20 m and 400 m, respectively, at the borders of the present study domain. Hence, SAETTA location errors are in the same order of magnitude as those of Meteorage CG location while the LMA should capture lightning in more detail than the LF LLS.

SAETTA data are employed for locations and times of coincident ISS-LIS or Meteorage observations. Therefore, they are analyzed in space and time regarding the detected ISS-LIS and Meteorage lightning activity. A combined space-time filter identifies SAETTA sources within 0.2° (both latitude and longitude) and (simultaneously) 0.3 s of corresponding ISS-LIS events and Meteorage pulses/strokes. The filtering per (ISS-LIS or Meteorage) flash allows for analyzing the concurrent VHF measurements with e.g. altitude information. Furthermore, SAETTA data are not used to exclude any ISS-LIS or Meteorage

observations and neither ISS-LIS nor Meteorage data are confined to any SAETTA data condition. They are, however, used to verify the applied data processing approaches, i.e. grouping elements (events, pulses/strokes) to flashes and the analysis of possible false alarms within the lightning detection of ISS-LIS and Meteorage. The maximum altitude of SAETTA sources is bounded at 15.0 kmASL and the maximum $reduced \ \chi^2$, which defines a measure for the overall uncertainty of the time-of-arrival based system (Thomas et al., 2004), is set to 0.5.





## 2.4  Flash - Grouping algorithm

The NASA LIS algorithm distinguishes events, groups and flashes (section 2.1). The distance between two events of one flash (Christian et al., 2000) can be up to 14.3 km at the edge of the FOV (up to 11.9 km nadir). The time constraint, $dt_{merge}$, confines the maximum time between two consecutive groups of the same flash as 330 ms. One flash cannot last longer than 2.0 s. An analysis of the P0.2 NASA flash sorting algorithm revealed that it tends to separate flashes when compared to concurrent SAETTA observations. Similar results were observed by Defer et al. (2005). Consequently, a new algorithm is developed to merge the ISS-LIS events to flashes. It has the additional advantage of treating both ISS-LIS events and Meteorage pulses/strokes. The fundamental elements sensed by each LLS, that are the smallest available lightning signals (events and pulses/strokes), are merged into flashes. More explicitly, an event of ISS-LIS (pulse/stroke of Meteorage) should belong to exactly one flash and a flash is defined as a group of events (pulses/strokes). Flash characteristics are derived from the underlying element characteristics, e.g. the positions of its elements are used instead of the mean flash location. This study makes use of the fundamental ISS-LIS event data as provided by the NASA prior to any data merging. It is accepted that ISS-LIS events do not have a direct representation in the Meteorage-like data. Former studies claimed that LIS groups roughly correspond to the physical processes detected by VLF/LF LLSs (e.g., Bitzer et al., 2016; Höller and Betz, 2010). Nevertheless, those studies found significantly more groups than pulses/strokes within the same region and time period. Bitzer et al. (2016) found for the number of TRMM-LIS groups to ENTLN pulses/strokes a factor of about 28.4 globally and even 3.7 in North America in 2013. Höller and Betz (2010) analyzed 6.7 groups per pulse/stroke on average. Due to those results, it is questionable whether LIS groups really correspond to (V)LF pulses/strokes. In addition, the detected lightning sources of the applied VHF LLS comply more with the LIS events than the groups. Using events rather than group centroids improve in particular the finding of the coincident LMA data. The analysis of flash extents profits from the use of events in that the extent of an ISS-LIS flash corresponds to the full illuminated area rather than the ISS-LIS group centroid locations. The representation of the flash extent (density) will influence the future assimilation of lightning data in NWP models. A statistical analysis of (ISS-LIS) events and LF strokes/pulses will also be of interest for creating a proxy optical data set, e.g. for MTG-LI, derived from LF data.

Our *grouping algorithm* analyzes the element (events or pulses/strokes) and groups the elements based on their relative location and time of occurrence to each other. First, the spatial and temporal constraints, $ds_{merge}$ and $dt_{merge}$, for elements within one flash must be determined. Then, a combined space-time test merges the elements into flashes. It starts with the first available element (in the data of one LLS) and identifies all elements (of the same LLS' data) within the range of the constraints. Thereby, an element can only belong to the same flash if both the distance to the initial element is less than $ds_{merge}$ and the time difference is shorter than $dt_{merge}$. All elements identified for a flash (and the initial element) are classified as *used*. For each used element within a flash, the test is repeated until no *unused* element can be added to the flash. This step allows for considering the potentially increasing extent and duration of a flash when adding new elements. The algorithm continues until all elements are classified as used. In general, our algorithm does not limit the duration of a flash. The number of elements per flash remains also free to the algorithm.





The algorithm verification includes a sensitivity study for $ds_{merge}$ and $dt_{merge}$ (Figure 2) and a comparison to NASA's algorithm and concurrent SAETTA observations (Figure 3).

Figure 2 gives the number of flashes analyzed from all observations of the approximately one-year-period by using different $ds_{merge}$ (panels a and c) and $dt_{merge}$ (b, d) for ISS-LIS (a, b) and Meteorage (c, d). In general, as expected, smaller $ds_{merge}$ and $dt_{merge}$ increase the flash numbers because less individual elements are part of a given flash and thus more flashes exist for the same elements. LIS flash numbers range from 236 to 4567 for the $ds_{merge}$ ($dt_{merge}$) between 50.0 km and 1.0 km (1.0 s and 0.1 s), respectively. For the same constraints, Meteorage flash numbers vary from 340 to 1720 flashes.

The ISS-LIS flash number decreases rapidly for $ds_{merge}$ between 0 km and 10 km (Figure 2(a)). The rapid change depends on the pixel size within the ISS-LIS image. Hence, it is expected that events of one flash are partitioned within the same frame if the $ds_{merge}$ becomes smaller than the ISS-LIS image pixel size. ISS-LIS flash numbers remain constant for $ds_{merge}$ greater than 15 km for all tested $dt_{merge}$. 0.3 s balances the need of consistency and the wish for a strict $dt_{merge}$ (Figure 2(b)). The resulting flashes are verified against concurrent 3D SAETTA sources which supported the choice of our constraints and the

identification of resulting flashes. Both chosen constraints for ISS-LIS flashes, 15 km for $ds_{merge}$ and 0.3 s for $dt_{merge}$, are similar to the P0.2 NASA flash sorting algorithm.

The same algorithm is applied to group the Meteorage pulses/strokes into flashes. It needs, however, modified constraints $ds_{merge}$ and $dt_{merge}$ since Meteorage pulses/strokes do not always represent the same physical processes as ISS-LIS events and occur with significantly lower counts. Meteorage pulses/strokes do not always cover the structure and duration of a lightning

flash. Figure 2(b, d) is analyzed for Meteorage flash numbers as demonstrated for ISS-LIS flash numbers in (a, c). To find constant constraints suitable for various situations (e.g. vertical cloud structures, severity of a storm, flash rate), resulting flashes for different constraints are verified manually using SAETTA observations. Conclusively, Meteorage pulses/strokes belong to the same flash if they are detected within 20 km and 0.4 s. Due to the limited number of pulses/strokes, Meteorage $ds_{merge}$ and $dt_{merge}$ are coarser than the ISS-LIS merging constraints. Our constraints (20 km, 0.4 s) are consistent with the

Meteorage-own flash grouping algorithm using a separation distance of less than 10 km for subsequent CG strokes and 20 km if IC pulses are involved. The delay between supsequent discharges of the same flash must be smaller than 0.5 s in Meteorage's algorithm. Höller and Betz (2010) provided a clustering of LINET VLF/LF pulses/strokes to flash scale with 10 km and 1.0 s in space and time, respectively. Hence, their merging constraints for a flash are finer in space but coarser in time.

One must mention, though, that the determination of $ds_{merge}$ and $dt_{merge}$ does not ensure a perfect arrangement of the

elements in flashes. The objective is to find constraints leading to statistical representations of flashes in the ISS-LIS and Meteorage data. Therefore, all identified flashes are double-checked against concurrent 3D SAETTA observations. Even if it is sometimes challenging to separate the flashes in the SAETTA data, the detailed VHF mapping helps to understand the processes leading to the identification of the ISS-LIS and Meteorage flashes. The SAETTA data are also used to find possible false alarms in the ISS-LIS and Meteorage.

Figure 3 demonstrates the behavior of NASA's flash merging algorithm and our developed algorithm for one example. It shows a short time period (6.5 s) during one ISS overpass on September 10, 2017. In Figure 3(a), there is the map of flash locations from the P0.2 NASA flash merging algorithm and our developed algorithm as well as concurrent VHF SAETTA





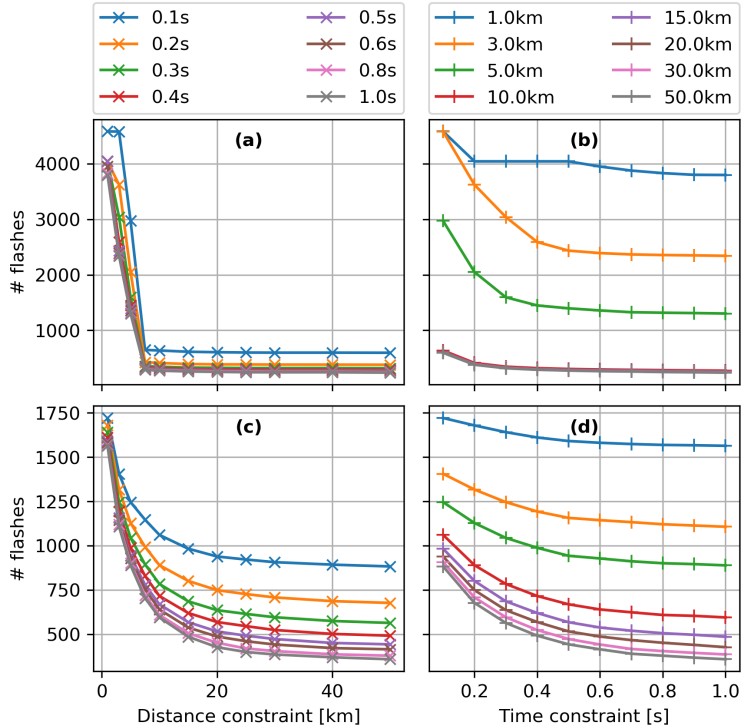

**Figure 2.** Total flash number based on constant, equal time constraint $dt_{merge}$ (line color) with varying distance $ds_{merge}$ for elements of ISS-LIS (a) and Meteorage (c) flashes. (b) and (d) with constant, equal distance $ds_{merge}$ (line color) and varying $dt_{merge}$ for ISS-LIS and Meteorage, respectively.

sources. The ISS-LIS events (not plotted) coincide in general well with the SAETTA observations in both location and time. The mapped observations are presented in latitude, longitude and altitude time series in Figure 3(b). 20 flashes from NASA's

algorithm are confronted with 11 flashes from our developed algorithm for the same ISS-LIS events. NASA's merging algorithm somehow splits some flashes, e.g. the flash between 5.3 s and 6.2 s where NASA's algorithm identifies 5 flashes. Our algorithm finds a single flash for that period and the concurrent SAETTA observations support this result.

Our developed algorithm was in addition tested versus GLM flash-scale data using the distance between events and not (as ISS-LIS) between group centroids in order to merge events/groups to flashes. Time and spacing for events/groups of one GLM

flash are 330 ms and 16.5 km, respectively (Goodman et al., 2013). The GLM flash-scale data agree very well with the flashes identified by our algorithm from the underlying GLM events.

## 2.5 Flash - Matching algorithm

ISS-LIS and Meteorage detect flashes in a different way. It was described how the different signals can be merged into a common entity, namely a flash. Our intercomparison of LLSs uses flash-scale to find concurrent observations. Our algorithm





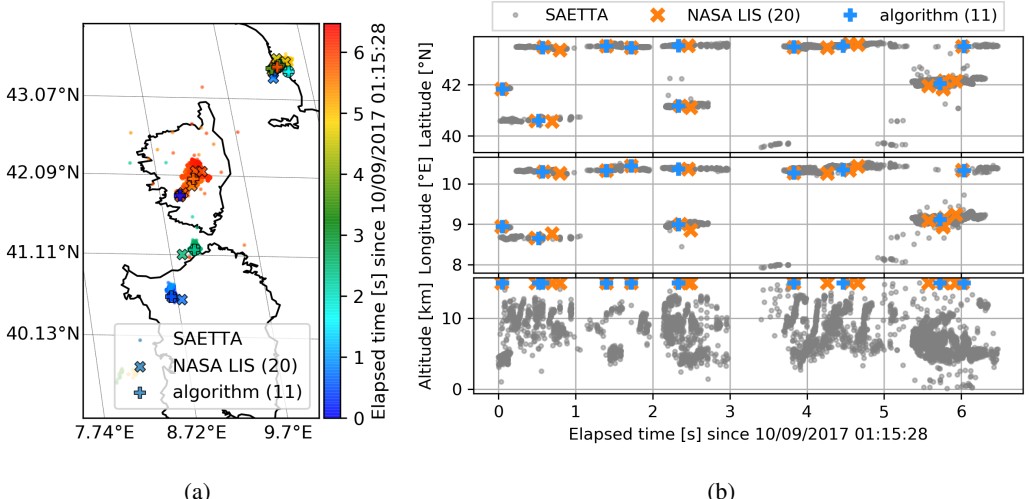

(a)          (b)

**Figure 3.** The map (a) and time series (b) of SAETTA observations and the mean flash positions based on the NASA LIS algorithm and our developed algorithm for one situation on Sept. 10, 2017. The numbers in parentheses in the legend indicate the number of identified flashes (both algorithms analyzed the same LIS events). Colors in (a) represent the elapsed time from the initial lightning activity. Note: Flash altitudes in (b) are not known for ISS-LIS flashes but plotted at 15 km.

scans the individual flashes of both LLSs and sorts them into one of the four following categories: *LIS detected by both* (i.e. an ISS-LIS flash has a coincident Meteorage flash), *LIS-only* (i.e. no coincident Meteorage observations), *Meteorage detected by both* (i.e. a Meteorage flash with concurrent ISS-LIS events) or *Meteorage-only* (i.e. ISS-LIS does not detect the flash). Matching criteria in space ($ds_{match}$) and time ($dt_{match}$) are specified. The criteria $ds_{match}$ and $dt_{match}$ do not address the flash mean position and time, respectively, but the single events or pulses/strokes within a flash. Flashes detected by one LLS

are characterized as *matched* to a given flash of the second LLS if they occur within the range of $ds_{match}$ and $dt_{match}$ to the elements of the given flash. As a given flash detected by a given LLS does not necessarily correspond to exactly one matched flash, the two categories *LIS detected by both* and *Meteorage detected by both* are expected to have different counts. It is also possible that a flash fulfills the matching criteria of more than one given flash and thus is collocated to more than one flash.

  The analysis of distances and timing offsets between matched flashes refines the results further: The closest flash detected

by the second LLS is identified for a given flash of the given LLS. Our algorithm analyses the underlying elements of each flash. The algorithm starts with one percent of both $ds_{match}$ and $dt_{match}$ seeking for any element detected by the second LLS around any element of the given flash. The allowed distance and time difference increase iteratively by one percent of $ds_{match}$ and $dt_{match}$, respectively, for a given flash (and all its elements) until a match is found. If the allowed distance (timing offset) exceeds $ds_{match}$ ($dt_{match}$), the algorithm stops and the flash is called *unmatched* (Note: The refined analysis is performed for

matched flashes only, however, the algorithm can treat the unmatched flashes, too.). One or more matches for the given flash





are possible because of the discrete increments from one iteration to the following. There might also be flashes within similar distance and similar time offset to the given flash.

The criteria $ds_{match}$ and $dt_{match}$ are determined through a sensitivity study of the relative DEs of ISS-LIS and Meteorage (Figure 4). A spatial criterion lower than 10.0 km reduces the relative DE of both ISS-LIS and Meteorage rapidly (Figure 4(a for ISS-LIS, c for Meteorage)). In general ISS-LIS' relative DE is more sensitive to both $ds_{match}$ and $dt_{match}$ than Meteorage relative DE. This result is triggered by the low number of Meteorage pulses/strokes (compared to the number of ISS-LIS events) hampering effectively the finding of suitable elements, i.e. pulses/strokes, for a collocation. ISS-LIS relative DE decreases within the entire range of investigated times $dt_{match}$. The most sensitive behavior occurs for $dt_{match}$ up to 1.5 s (Figure 4(b)). Meteorage appears to be sensitive to $dt_{match}$ onl yup to 0.5 s (Figure 4(d)). Despite the differences in sensitivity to the criteria between ISS-LIS and Meteorage, it is aimed at using the same $ds_{match}$ and $dt_{match}$ for both LLSs. Finally, $ds_{match}$ of 20 km and $dt_{match}$ of 1.0 s are chosen to balance the individual sensitivities of the LLSs to the criteria. They allow to identify matches if, for example, ISS-LIS detects primary IC discharges of a flash and Meteorage only detects a CG stroke occuring during the final stage of the same flash. Our criteria are relatively coarse compared to some former studies (section 1). Höller and Betz (2010) applied the same $dt_{match}$ but an even coarser $ds_{match}$ (i.e. 30 km) to match LINET VLF/LF flashes and TRMM-LIS flashes. Further investigation of the matched flashes, e.g. the distributions of the distances and timing offsets, will demonstrate to which extent matches rely on the fairly coarse criteria.

## 3    Results

The different LLSs detect flashes in different ways and with distinct characteristics. In this section, flash observations are compared and analyzed. As an example, the ISS overpass with the corresponding observations of ISS-LIS, Meteorage and SAETTA in Figure 1 comprises (almost) the entire study region. It lasted 169 seconds, from FOV entering to leaving the region. The effective viewtime per 0.5° x 0.5° grid box is indicated in grayscale in Figure 1(a). Wide parts of the domain have been seen for at least 60 seconds. Figure 1(b) shows additionally an IR satellite image indicating the cloud tops. The example of a single flash observed by all three LLSs during this overpass is given in Figure 1(c). SAETTA captures the most detail of the flash structure and there are significantly more ISS-LIS events than Meteorage pulses/strokes. All but the first Meteorage signals indicate an IC pulse. Since the first stroke is of type CG, the entire flash is characterized as CG-flash.

First, relative DEs of ISS-LIS and Meteorage are elucidated. The comparisons of matched flash locating and timing differences are discussed and finally characteristics of flashes, with special interest in differences between matched and unmatched flashes, are analyzed.

### 3.1    Detection comparison

Our DE analysis is realized on the flash scale. Flashes were preliminarily identified by our in-house algorithm, which merges ISS-LIS events and Meteorage pulses/strokes according to their locations and times of occurrence. Further investigations break the flash scale down into events and pulses/strokes, e.g. for the flash characteristics.



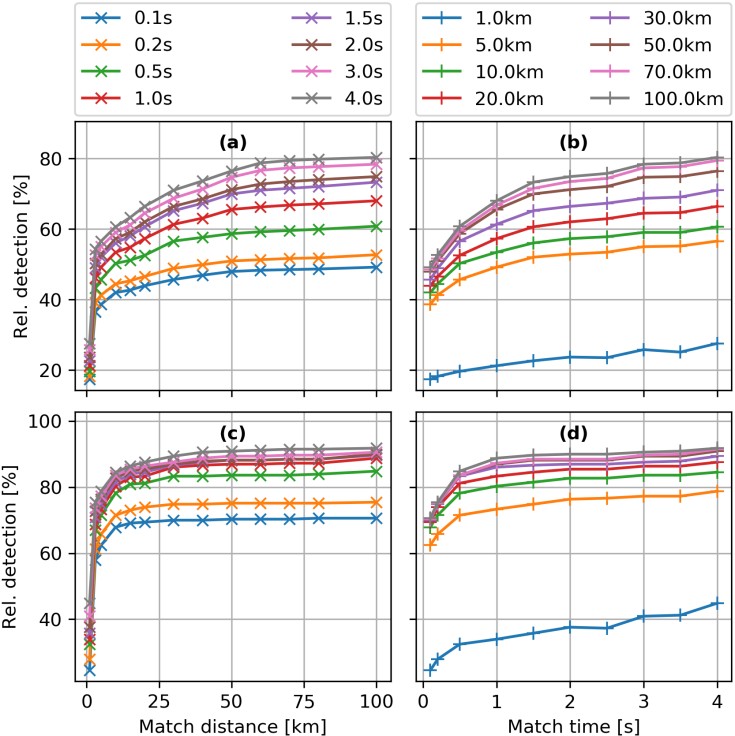

**Figure 4.** Relative detection efficiency based on constant, equal time criterion $dt_{match}$ (line color) with varying distance $ds_{match}$ for ISS-LIS (a) and Meteorage (c). (b) and (d) with constant, equal distance $ds_{match}$ (line color) and varying $dt_{match}$ for ISS-LIS and Meteorage, respectively.

The period of observations spans from March 01, 2017 to March 20, 2018. In total, 330 ISS-LIS flashes and 569 Meteorage flashes are identified by our algorithm.

325      Besides the DE, the Probability of False Alarm (POFA) characterizes the quality of detection. Quantifying the POFA needs knowledge about the truth, that is the real number of flashes. SAETTA could provide a reference value to quantify the POFA, however, not all stations have operated continuously for the entire study period. Signals from at least six stations are needed to reconstruct and locate a discharge signal. 31 of 330 (89 of 569) ISS-LIS (Meteorage) flashes are not detected by SAETTA. SAETTA's detection efficiency and accuracy decreases also with the distance to the network's center. A vast majority of more

330   than 90 % of all flashes occurred outside a distance of 100 km from SAETTA's center. Only 3 (12) of the ISS-LIS (Meteorage) flashes missed by SAETTA are located within 100 km of SAETTA's center. Due to the low total flash number within this close domain to SAETTA, a statistical analysis is ambiguous. Pédeboy et al. (2018b) reported Meteorage flashes missed by SAETTA with (absolute) peak currents exceeding 100 kA. Two of the twelve missed Meteorage flashes close to SAETTA exhibit an (absolute) current above 100 kA. SAETTA data can in fact not serve for the desired true flash numbers and the

335   POFA of Meteorage and ISS-LIS cannot be calculated.





**Table 1.** Relative detection efficiencies (DEs) of Meteorage and ISS-LIS. The values in parentheses give the relative DEs for flashes with at least two elements. The flash numbers (100 %) to calculate the DEs are indicated. Note: ISS-LIS (Meteorage) DE uses Meteorage (ISS-LIS) flash numbers.

|  | Overall | Daytime | Nighttime | IC-flash | CG-flash |
|---|---|---|---|---|---|
| ISS-LIS DE [%] | 57.3 (62.4) | 53.9 (60.8) | 58.7 (63.0) | 59.3 (68.8) | 53.5 (55.4) |
| Meteorage flash number | 569 (367) | 167 (102) | 402 (265) | 369 (192) | 200 (175) |
| Meteorage DE [%] | 83.3 (83.9) | 80.0 (80.2) | 84.8 (85.5) | - | - |
| ISS-LIS flash number | 330 (316) | 100 (96) | 230 (220) | - | - |

It should be mentioned, however, that 60.7 % (54) of the Meteorage flashes without concurrent SAETTA sources contain one pulse/stroke only. Those flashes with only one pulse/stroke (or one event in case of ISS-LIS) are referred to as *single element flashes*. Missing SAETTA observations for a single element flash might be indicative of a locating and timing problem of the ISS-LIS event or Meteorage pulse/stroke (possible false alarm). Our DE analysis distinguishes results for the full data set and excluded single element flashes. 316 ISS-LIS and 367 Meteorage flashes remain after excluding the single element flashes. Thus, the ISS-LIS (Meteorage) flash number is reduced by 14 (202) flashes compared to the overall count. ISS-LIS single element flashes are rare while there is a significant amount of Meteorage single element flashes. The result is related to the differences in optical and LF lightning detection. The entire data set contains 16,881 ISS-LIS events and 2,144 Meteorage pulses/strokes (487 CG, 1657 IC). 15,578 events (92 %) are distributed over the ISS-LIS flashes with match. For Meteorage, 1,439 pulses/strokes (271 CG, 1168 IC) constitute the flashes with matches (67 %). Hence, 55.6 % (70.5 %) of the CG strokes (IC pulses) belong to flashes with coincident ground-space-detection. Despite coarser $ds_{merge}$ and $dt_{merge}$ for a Meteorage flash than an ISS-LIS flash, Meteorage observed 239 flashes more than ISS-LIS within similar regions and time frames.

Figure 5(a) presents a histogram of the total flash detection counts within the four categories introduced in section 2.5. The number of single element flashes is marked. Additionally, it includes a map of the locations of the flashes within each category. Figure 5(b) maps the flashes as 2D-histogram on a 0.1° x 0.1° grid. Flashes are detected all over the study domain for both ISS-LIS and Meteorage without any apparent pattern even if the number of flashes is not sufficient to be statistically representative.

Table 1 summarizes all relative DEs. Daytime covers the time from 05:00 UTC to 17:00 UTC. Nighttime flashes are defined between 17:00 UTC and 05:00 UTC.

ISS-LIS was able to detect 326 of the 569 recorded Meteorage flashes from March 01, 2017 to March 20, 2018, a relative DE of 57.3 %. If the notable number of Meteorage single element flashes is neglected, ISS-LIS detected 229 of the remaining 367 Meteorage flashes (62.4 %). ISS-LIS shows a low relative DE of less than 54 % for daytime flashes. 58.7 % of the Meteorage nighttime flashes are detected by ISS-LIS. In particular the nighttime relative DE cannot reach literature expectations of over 90 % for LIS (Boccippio et al., 2002). ISS-LIS' relative DE does significantly depend on the Meteorage flash type. A flash with at least one CG stroke, referred to as CG-flash, is detected in only 53.5 % of the cases while a pure IC-flash is detected with 59.3 % relative DE. ISS-LIS could detect 68.8 % of the occurring Meteorage IC flashes with at least two pulses. If flashes





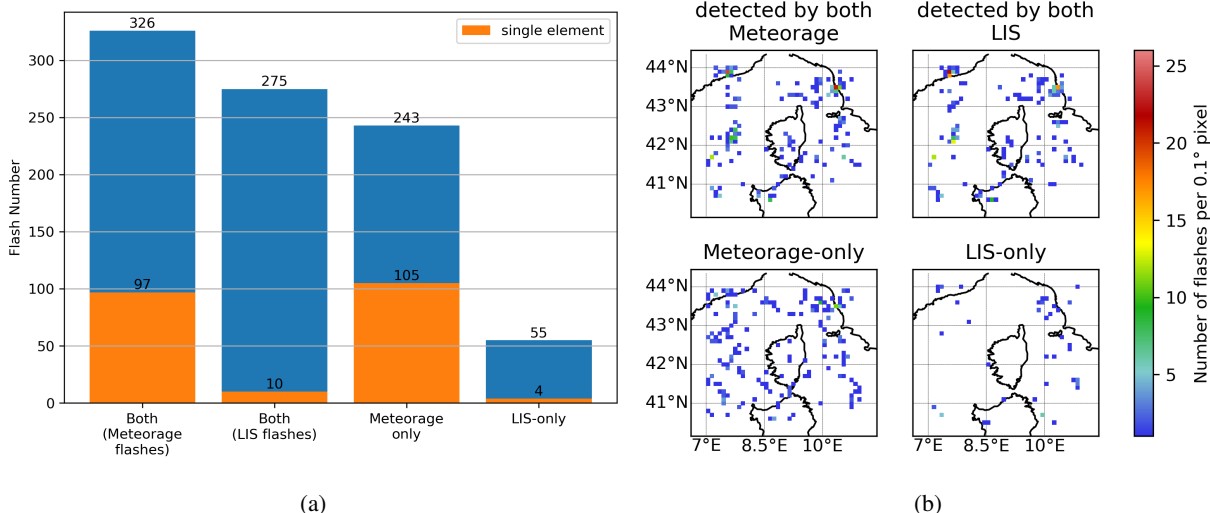

(a)  (b)

**Figure 5.** Flash category histogram (a) and spatial distribution (b) while matching ISS-LIS and Meteorage flashes for the available ISS overpasses from March 2017 to March 2018. Categories show the number of Meteorage flashes seen by ISS-LIS [Both (Meteorage flashes)], the number of ISS-LIS flashes detected also by Meteorage [Both (LIS flashes)] and flashes detected either by Meteorage [Meteorage-only] or ISS-LIS [LIS-only]. The numbers of single element flashes (event for ISS-LIS, pulse/stroke for Meteorage) are marked for each category as indicated.

with at least two pulses/strokes are considered, the relative DE of IC flashes surpasses that of CG-flashes by almost 14 % and increases compared to the total IC flash relative DE by 9.5 %. Hence, especially CG-flashes and single pulse IC-flashes decrease the total DE of ISS-LIS. All relative DEs use $ds_{match}$ of 20 km and $dt_{match}$ of 1.0 s. Finer criteria would further
decrease the relative DE of ISS-LIS (higher sensitivity to the criteria than Meteorage).

Out of the total 330 ISS-LIS flashes, Meteorage detected 275 (83.3 %). The DE of Meteorage relative to ISS-LIS flashes with at least two events equals 83.9 % (265 of 316 flashes). The relative DE of the VLF/LF Meteorage LLS appears to be significantly higher than in former studies (section 1) using LF LLSs and TRMM-LIS. It is assumed that the ISS-LIS detection efficiency is similar to that of TRMM-LIS in general (personal communication, R. Blakeslee) and thus Meteorage provides a
high quality LF LLS. Moreover, especially Meteorage detection efficiency appears to be quite resistant to changes of $ds_{match}$ and $dt_{match}$. For example, halving both criteria (10 km in space, 0.5 s in time) results in a relative detection efficiency of about 78 %. More details about the sensitivity to the matching criteria can be found in Section 2.5.

Meteorage detected 80.0 % of the 100 ISS-LIS daytime flashes. Its relative DE reaches 84.8 % for 230 ISS-LIS nighttime flashes. The relative DE depends on both the performance of the LLS itself but also the performance and locating accuracy
relative to the reference LLS. As ISS-LIS detects flashes optically, the influence of different lighting on ISS-LIS daytime and nighttime accuracy is investigated as part of the following section 3.2.



### 3.2 Distances and timing offsets between collocated flashes

In this section, the matched ISS-LIS and Meteorage flashes are studied regarding their relative location and time of occurrence. For each element of a flash detected by one LLS the closest (in time or in space, not a combined filter here) element of the

matched flash(es) accounts for the statistic. One element can be closest to multiple elements of the second LLS. The entirety of elements of flashes with matches is analyzed statistically. Figure 6 presents the results for distances (a) and timing offsets (b) between events and pulses/strokes.

Figure 6(a) shows histograms of the distance between a given ISS-LIS event (Meteorage pulse/stroke) and the closest pulse/stroke (event) of a matched flash. The distribution given an ISS-LIS event peaks primarily between 2.50 km and 3.00 km

and secondarily at about 4.50 km with a median (mean) of 4.74 km (5.68 km). The distribution given a Meteorage pulse/stroke has a broad maximum from 0.75 km to 2.75 km with a median (mean) of 2.31 km (3.60 km). The Meteorage pulse/stroke distance distribution features a more pronounced (if wider) peak for less distance than the distribution given an ISS-LIS event. This is due to the calculation method and the numbers of available events and pulses/strokes. The higher number of (and smaller distance between) ISS-LIS events allows in general for finding a closer event to a given Meteorage pulse/stroke than vice versa.

The cumulative distribution functions (CDFs) within the plotted interval (Figure 6(a)(ii)) show that the distance distribution given an ISS-LIS events has a larger tail than the distribution given a Meteorage pulse/stroke. The 60th percentile is found at approximately 5.5 km and 2.6 km for a given ISS-LIS event and Meteorage pulse/stroke, respectively. Both Meteorage IC pulses and CG strokes exhibit similar distributions to the overall Meteorage pulses/strokes (also in the CDFs, Figure 6(a)(ii)) with the peak between 0.75 km and 2.75 km. The median (mean) distance for IC pulses and CG strokes and their match equals

2.36 km (3.63 km) and 2.22 km (3.51 km), respectively. Hence, CG strokes feature a slightly lower distance to matched events than IC pulses.

Distances are calculated between the closest ISS-LIS events (not groups or flashes as in former studies) and LF pulses/strokes. The group-pulse/stroke distances should be equal or greater than event-pulse/stroke distances as events provide a finer resolution of the lightning discharge. Bitzer et al. (2016), who used TRMM-LIS groups and ENTLN pulses, found for both

conditional distributions median (mean) differences in location between 7.0 km and 7.2 km (7.6 km and 7.9 km) in North America. Those values are in accordance with the distances observed by Rudlosky et al. (2017) between TRMM-LIS flashes and GLD360 strokes. All their results show a similar order of magnitude to our findings comparing ISS-LIS events and Meteorage LF pulses/strokes.

The optical ISS-LIS sensor might be affected by different lighting. Therefore, the accuracy of ISS-LIS flashes relative

to ground-based LLSs is explicitly investigated during day and night (not shown as Figure). Daytime flash distances are concentrated mainly between 2.0 km and 5.0 km and the distribution peaks at about 3.5 km. The ISS-LIS nighttime flash distribution peaks at about 5.5 km distance to matched Meteorage flashes. Given an ISS-LIS flash, the CDF distribution also rises faster for daytime than for nighttime flash distances. Hence, distances between coincident flashes are in fact smaller during daytime than during nighttime. The comparison of ISS-LIS flashes to SAETTA reveals a small difference of up to 0.05° latitude

and longitude during both day- and nighttime. ISS-LIS flashes tend to occur slightly south and west of the corresponding





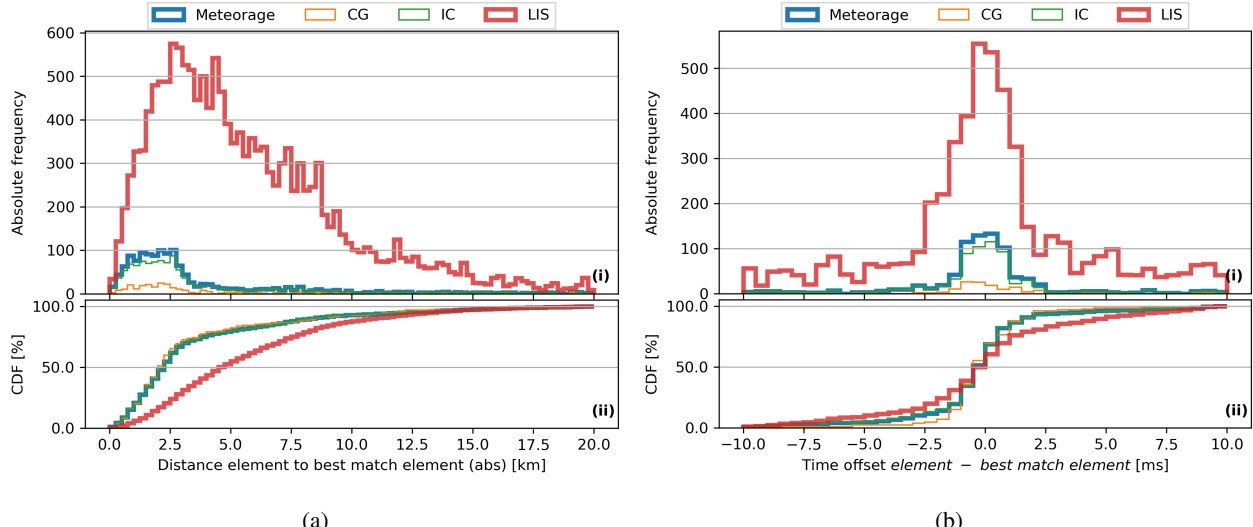

(a)                                                                  (b)

**Figure 6.** Best match distance (a) and time offset (b) between a given Meteorage pulse/stroke (LIS event) and the closest ISS-LIS event (Meteorage pulse/stroke). Histogram (i) and cumulative distribution function (ii) with bins of 0.25 km (a) and 0.5 ms (b). Analyzed pulses/strokes (events) belong to flashes with matches (spatial and temporal filters) while the pulses/strokes (events) of unmatched flashes are not considered. For Meteorage, the discharge types [CG, IC] are distinguished. Only elements with absolute timing offsets of less than 10 ms are included in the plotted time offset distribution. A positive time indicates the given element occurred later than the best match.

SAETTA observations. The small locating difference, considering $ds_{match}$ of 20 km and ISS-LIS spatial resolution of 4.5 km (nadir), does not significantly influence our results. In particular, ISS-LIS maintains its locating accuracy during daytime and during nighttime.

The timing offset subtracts the time of the matched element from the time of the given element. It yields positive and negative values according to which element occurred first, with a positive value indicating that the given element occurred later than its match. Again, the two conditions *given an ISS-LIS event* and *given a Meteorage pulse/stroke* are applied. The resulting distribution (Figure 6(b)) peaks between -0.5 ms and 0.5 ms for a given ISS-LIS event and between -1.0 ms and 1.0 ms for a given Meteorage pulse/stroke. The tails of the distribution, with absolute timing offset longer than 10 ms and up to 1.0 s, are not plotted. They are larger for a given ISS-LIS event than for a given Meteorage pulse/stroke. It is observed that Meteorage

pulses/strokes often do not cover the entire duration of a flash. ISS-LIS events reflect the actual flash duration (reference to concurrent SAETTA sources) better than the Meteorage pulses/strokes. Hence, given an ISS-LIS event and looking for a matched Meteorage pulse/stroke, the number of available pulses/strokes is often limited. Several events can have the same closest pulse/stroke even if the events occurred in different time frames. It increases the probability of larger timing offsets especially for a given ISS-LIS event compared to a given Meteorage pulse/stroke. The CDFs (Figure 6(b)(ii)) reveal that about

20 % (5 %) of the ISS-LIS events (Meteorage pulses/strokes) shown here exhibit timing offsets of more than positive 2.5 ms. About 20 % (10 %) of ISS-LIS events (Meteorage pulses/strokes) have values lower than -2.5 ms. In the overall distribution





(not shown), time offset exceed positive 10.0 ms for 43 % (22 %) of ISS-LIS events (Meteorage pulses/strokes) and are more negative than -10.0 ms for 25 % (22 %) of ISS-LIS events (Meteorage pulses/strokes). The distribution given an ISS-LIS event is slightly skewed towards positve time offsets (given ISS-LIS event occurred later than its best match stroke/pulse).

The overall median (mean) values yield 2.36 ms (54.60 ms) and -0.00 ms (2.70 ms) given an ISS-LIS event and Meteorage pulse/stroke, respectively. The large mean value for a given ISS-LIS event is an artifact of skewed distribution. Considering the ISS-LIS integration frame time of 2.0 ms, the remaining average statistics are close to the temporal accuracy of ISS-LIS. Both conditional distributions given ISS-LIS and given Meteorage show an overall similar shape (also Figure 6(b)(ii)). The matched element, considering both the ISS-LIS and Meteorage distributions, occurs with similar probability earlier or later (or

simultaneously) than the element itself and the distribution peak is centered at zero time offset. This is an interesting finding since e.g. Höller and Betz (2010) and Bitzer et al. (2016) found that TRMM-LIS detected lightning on average one to two milliseconds later than the ground-based LLSs. This is not generally the case for ISS-LIS in our study (and again one must consider the ISS-LIS integration time frame of 2.0 ms). Although the order of magnitude of the time offsets agrees well with our results. Timing differences can in fact be directly compared to those studies as the closest event provides the same time as

the closest group (groups merge several events within the same time frame and in adjacent pixels of ISS-LIS).

The distribution given an IC pulse is also symmetric around zero and shows a maximum between -1.0 ms and 1.0 ms (Figure 6(b)). Its median (mean) is 0.00 ms (4.29 ms). For CG strokes, however, the distribution maximum reaches from -1.0 ms to 0.0 ms. The negative median (mean) of -0.07 ms (-4.32 ms) indicates that ISS-LIS detected CG lightning slightly later than Meteorage. It might account for the time the light of the CG lightning needs to propagate towards the higher parts of

the cloud and to become visible from space.

### 3.3 Characteristics of detected flashes

The previous sections dealt with relative DEs, location and times of coincident ISS-LIS and Meteorage records. In this section the unmatched flashes (42.7 % Meteorage, 16.7 % ISS-LIS) are also considered to investigate the following flash characteristics: the number of elements (events, pulses/strokes) per flash, flash extent, flash duration, flash mean absolute (pulse/stroke)

amplitudes and individual pulse/stroke amplitudes, flash mean (event) radiance, flash maximum (event) radiance and the flash mean, minimum and maximum altitudes based on SAETTA observations. They are separated per matched and unmatched flash, per daytime (05:00 UTC to 17:00 UTC) and nighttime (17:00 UTC to 05:00 UTC) as well as per IC and CG. The ISS-LIS flash IC or CG attribute depends on the type of the matched Meteorage flash. There is no flash type associated with ISS-LIS-only flashes. As explained in section 2.4, ISS-LIS events are analyzed. The statistical results obtained would be similar using groups

instead of events, except for the flash extent and maximum radiance per flash. It should be mentioned that especially the number of daytime CG flashes is very limited (24 ISS-LIS, 42 Meteorage meaning <10 %). Flash extents add the north-south and the east-west distance of a flash. The north-south distance uses the maximum and minimum latitude of the flash elements. The east-west distance of a flash is defined as the distance between the longitudinal maximum and minimum of the elements at the mean latitude (as that distance depends also on the latitude). Flash durations, or the times from the first to the last element of a

flash, are in general not limited in this study.

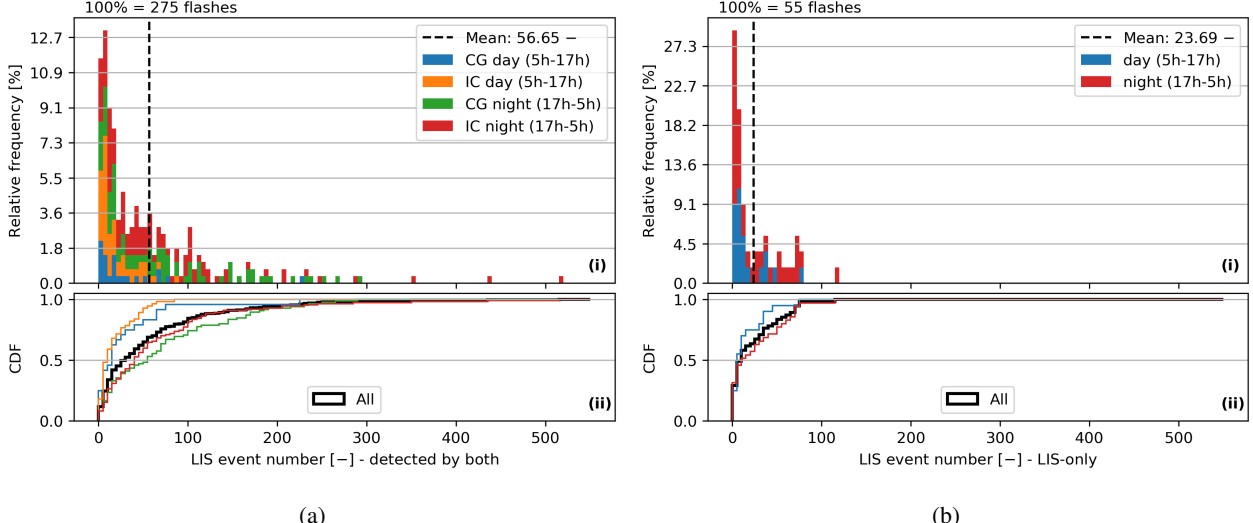

**Figure 7.** LIS event numbers of ISS-LIS flashes with coincident Meteorage flash (a) and unmatched ISS-LIS flashes (b). Daytime, nighttime and flash type, IC and CG, are indicated by the colors. Histogram (i) and corresponding CDF (ii) use the same colors. The CDF shows in addition a black curve for all data. The histogram bin width is constant at 5 events. Note: The CG/IC attribute for ISS-LIS flash needs the matched Meteorage flash and does not exist for ISS-LIS-only flashes. The mean value is plotted as dashed line. The total number of flashes is indicate above the histogram.

ISS-LIS flashes have on average 51.2 events (minimum 1, maximum 518). The flashes with a coincident Meteorage flash contain 56.7 events (minimum 1, maximum 518), while the ISS-LIS flashes without a match show only 23.7 events on average (minimum 1, maximum 116). About half (25 %) of the unmatched (matched) ISS-LIS flashes exhibit less than 10.0 events and about half (30 %) of those flashes were recorded during one single ISS-LIS frame. The event number distributions for

matched and unmatched ISS-LIS flashes are shown in Figure 7(a) and (b), respectively. It includes the histogram (i) and the CDF (ii). Daytime and nighttime flashes are distinguished for the flash types (only for matched flashes). The histogram's bars add the numbers of the different categories for the corresponding bin. All following Figures make use of the same layout. ISS-LIS nighttime flashes have about two times more events than daytime flashes. The background subtraction threshold for the optical signal is usually greater during daytime than during nighttime (difference of about 1.0 $\mu J \cdot sr^{-1} m^{-2} \mu m^{-1}$, overall

minimum event radiance at 9.0 $\mu J \cdot sr^{-1} m^{-2} \mu m^{-1}$) and can influence the number of events per flash with a relative reduction of event numbers on bright backgrounds (daytime) compared to dark backgrounds (nighttime). Additionally, ISS-LIS CG flashes comprise on average approximately 11 % more events than IC flashes.

The average number of events in matched ISS-LIS flashes is more than two times higher than in unmatched flashes. Accordingly, the matched ISS-LIS flashes feature also a larger extent and longer duration than the unmatched ISS-LIS flashes. Figures

8 and 9 present the results for the ISS-LIS flash extents and durations. ISS-LIS flash extents range from 0 km (single events) to 92 km. The average flash extents equal 19.8 km and 29.5 km for unmatched and matched ISS-LIS flashes, respectively.





Peterson et al. (2018), who studied the evolution and structure of extreme flashes observed by TRMM-LIS, found a LIS flash with maximum event separation of 162 km. The size results likely from an elongation due to scattering of optically bright discharges.

Here, it is observed that almost all flashes (except for 4) with an extent exceeding 40 km have coincident Meteorage flashes. ISS-LIS nighttime flashes are on average about 5 km larger than daytime flashes (and comprise more events). The result might indicate a better detection of dim events on very dark backgrounds during night compared to bright sunlit clouds at daytime. It could also result from an optical elongation of nighttime flashes. Large flashes with the maximum event separations in Peterson et al. (2018) also occurred at nighttime, but the groups of these flashes were not separated by a significant fraction of

the event separation. Fundamentally different cloud structures or types during day and night might also influence the results. It would need additional information, e.g. measuring infrared brightness temperatures for the cloud tops, to verify this hypothesis. Referring to the flash types, the mean extent of ISS-LIS CG flashes is about 5 km longer than for ISS-LIS IC flashes, however, the longest ISS-LIS flash is a nighttime IC flash.

One observed ISS-LIS flash lasted about 1.7 s (a CG nighttime flash), the longest duration found in this study. Peterson

et al. (2018) found spurious flash durations up to 28 s in convective clouds, which result from high flash rates and slow storm motion. One large propagating flash lasted 5.04 s in their study. Matched ISS-LIS flashes last on average almost twice as long as ISS-LIS-only flashes, i.e. 0.35 s versus 0.20 s (Figure 9). Long lasting flashes (duration longer than 0.5 s) have a high probability (92.6 %) of being detected by both LLSs. ISS-LIS nighttime flashes last statistically 0.1 s longer than the daytime flashes. The result accords with the higher relative DE of ISS-LIS and more detected events during the night than during the

day. The ISS-LIS flash duration distribution does not significantly depend on the flash type.

Meteorage flashes contain between 1 and 54 pulses/strokes (overall average 3.8). The distributions of pulse/stroke numbers per matched and unmatched flash are presented in Figure 10(a) and (b), including the stacked histogram (i) and the CDFs (ii) as explained for Figure 7. Meteorage flashes seen by ISS-LIS are composed of 4.4 pulses/strokes on average. Meteorage-only flashes contain 2.9 pulses/strokes on average. 97 of the 326 Meteorage flashes with coincident ISS-LIS flash have only one

pulse/stroke (10 CG, 87 IC). Among the 243 unmatched Meteorage flashes, 105 single pulse/stroke flashes are found (15 CG, 90 IC). As for ISS-LIS flashes, Meteorage flashes with match not only contain more pulses/strokes, but extend and last also longer than the unmatched flashes. The flash extent distributions in Figure 11(a) and (b) show a mean (maximum) of 12.1 km (147.5 km) and 6.9 km (109.2 km) for matched and unmatched flashes, respectively. ISS-LIS detected all IC Meteorage flashes with extents above 32 km. The longest flashes are categorized as CG nighttime. In general, Meteorage CG flashes extend further

than IC flashes. The mean extent equals 18.2 km (11.6 km) and 9.2 km (3.9 km) for matched (unmatched) CG and IC flashes, respectively. It is particularly small for unmatched IC flashes (as ISS-LIS can detect the longer IC flashes).

Meteorage flash durations support the findings, with matched flashes lasting on average (maximal) 0.22 s (2.3 s) and unmatched flashes lasting on average (maximal) 0.11 s (1.0 s). Figure 12(a) and (b) provide the duration distributions for Meteorage flashes. Distributions of both matched (a) and unmatched (b) flashes are sharply peaked for flashes shorter than 0.05 s (first

bin; including single element flashes, maximum of 13 pulses/strokes per flash). The CDF (Figure 12(a) and (b) (ii)) illustrates that Meteorage CG flashes (mean 0.28 s) last statistically longer than Meteorage IC flashes (mean 0.11 s).





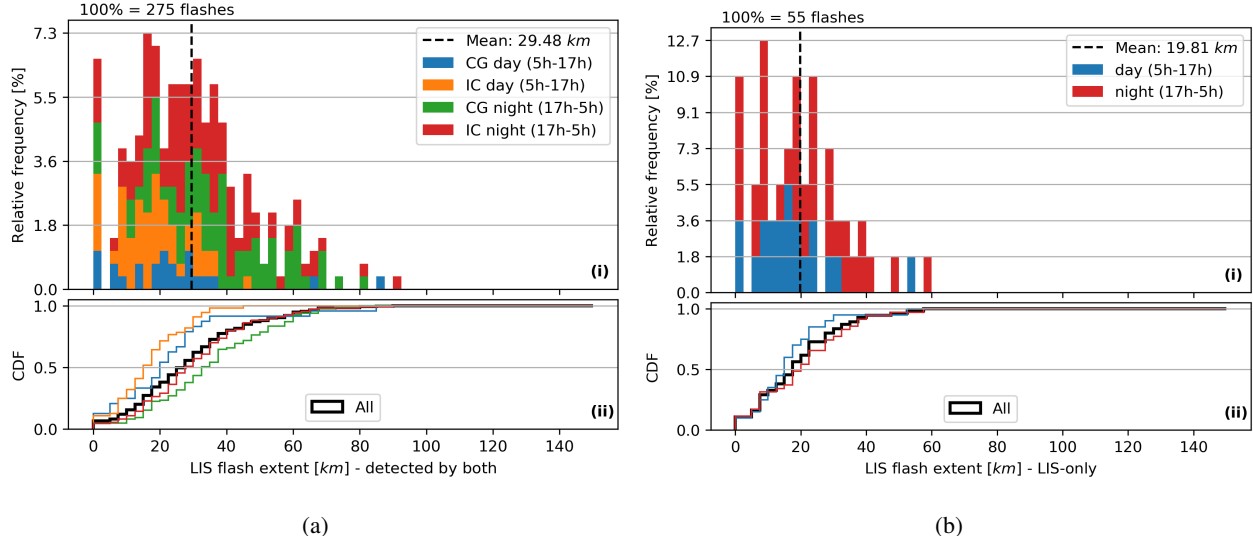

      (a)                                                 (b)

**Figure 8.** Flash extents of ISS-LIS flashes with coincident Meteorage flash (a) and unmatched ISS-LIS flashes (b). Daytime, nighttime and flash type, IC and CG, are indicated by the colors. Histogram (i) and corresponding CDF (ii) use the same colors. The CDF shows in addition a black curve for all data. The histogram bin width is constant at 2.5 km. Note: The CG/IC attribute for ISS-LIS flash needs the matched Meteorage flash and does not exist for ISS-LIS-only flashes. The mean value is plotted as dashed line. The total number of flashes is indicate above the histogram.

Seven (2 ISS-LIS, 5 Meteorage) exceptionally long flashes (extent > 90 km or duration > 1.5 s) are analyzed using concurrent SAETTA observations. The VHF sources highlight that there can be concurrent flashes that either merge and form one flash or propagate at different height levels. ISS-LIS and Meteorage detect both types as continuous flashes as the LLSs capture the

flashes two-dimensionally. They can in particular not distinguish the different altitudes for the latter.

Conclusively, matched flashes contain more elements, are more extended and last longer than unmatched flashes for both ISS-LIS and Meteorage records. Meteorage flashes appear to be on average both smaller in extent and shorter in duration than ISS-LIS flashes. The finding is in accordance with the different expectations on optical (LIS) and LF (Meteorage) signals.

Then, the VHF SAETTA LLS is used to determine the altitude range of each ISS-LIS and Meteorage flash. Three additional

flash characteristics are defined: flash mean altitude, flash minimum altitude and flash maximum altitude. The minimum altitude is defined as the 10th percentile of the altitudes of concurrent SAETTA sources rather than the true minimum. It is aimed at reducing the influence of noise in the data. In the same manner the flash maximum altitude equals the 90th percentile of the concurrent SAETTA source altitudes instead of the true distribution maximum. Since not every ISS-LIS and Meteorage flash happens within the SAETTA detection range, flash numbers are reduced compared to the ones discussed in section 3.1 to

256 ISS-LIS flashes with match, 43 ISS-LIS-only flashes, 292 Meteorage flashes with match and 188 Meteorage-only flashes. ISS-LIS mean event radiance and Meteorage mean pulse/stroke amplitude distributions are examined for the analyzed altitude levels of flashes. It should be mentioned that the lowest altitude of detectable VHF sources increases with distance to the LMA





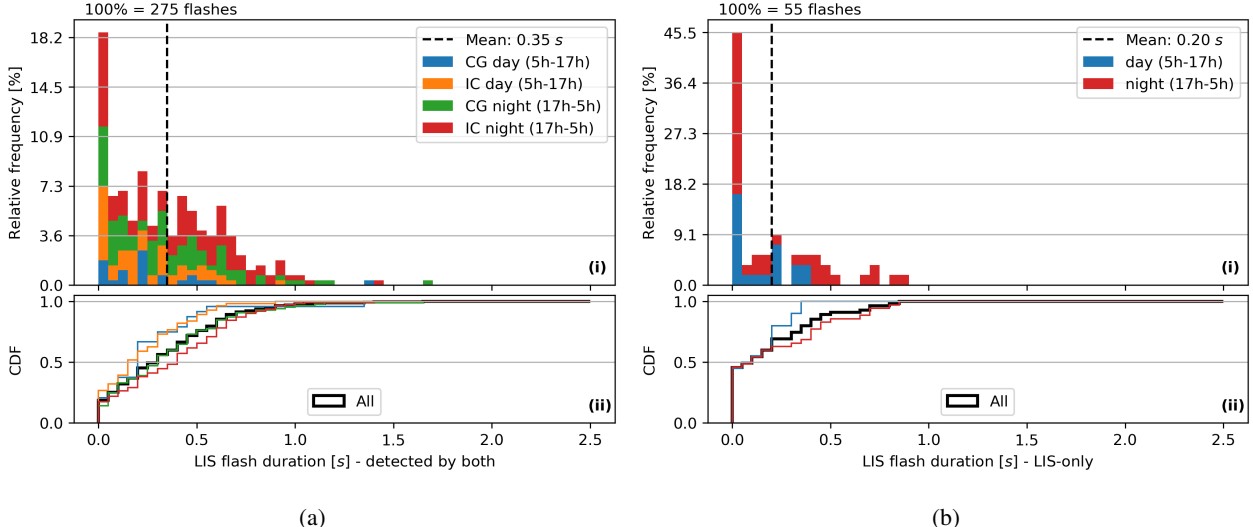

**Figure 9.** Flash duration of ISS-LIS flashes with coincident Meteorage flash (a) and unmatched ISS-LIS flashes (b). Daytime, nighttime and flash type, IC and CG, are indicated by the colors. Histogram (i) and corresponding CDF (ii) use the same colors. The CDF shows in addition a black curve for all data. The histogram bin width is constant at 0.05 s. Note: The CG/IC attribute for ISS-LIS flash needs the matched Meteorage flash and does not exist for ISS-LIS-only flashes. The mean value is plotted as dashed line. The total number of flashes is indicate above the histogram.

network mainly due to Earth's curvature and also due to shading by the relief, especially in the South of the domain (Coquillat et al., 2019). Hence, flash mean and especially minimum altitudes might suffer from undetected VHF sources in low altitudes.

Figure 13 presents the mean altitude of matched (a) and unmatched (b) ISS-LIS flashes as histograms (i) and CDFs (ii). The distribution of the flash mean radiances in each altitude bin is included as blue boxplot diagram (with mean marked as diamond, outliers not plotted). LIS flashes with coincident Meteorage flash have an average mean altitude of 8.2 km (Figure 13(a)(i)). The same average mean flash altitude is found for ISS-LIS-only flashes (Figure 13(b)(i)). The distribution of unmatched ISS-LIS flashes fits that of matched ISS-LIS flashes although the number of unmatched flashes is low.

The overall ISS-LIS flash mean (maximum) altitude distribution, that is dominated by 83.3 % flashes with match, peaks at about 9.5 km (11.0 km to 11.5 km), as shown in the histograms in in Figures 13(a)(i) and 14(a)(i). The daytime distribution has a second mode near 5.0 km (7.0 km) of altitude. About half of the flashes reach altitudes of 11.0 km and above (Figure 14(ii)), a noteworthy high value considering the tropopause in ten to twelve kilometers of altitude.

    Differences between matched and unmatched ISS-LIS minimum flash altitudes approximate 0.5 km, with matched flashes 540 showing lower minima (distributions not shown). The difference is significant as it exceeds the predicted SAETTA altitude error (about 0.2 km over wide parts of the domain). 89.7 % of the 126 ISS-LIS flashes with minima less (or equal) than 6.0 km of altitude have a coincident Meteorage flash. ISS-LIS flashes with minima above 6.0 km (173) are detected by Meteorage





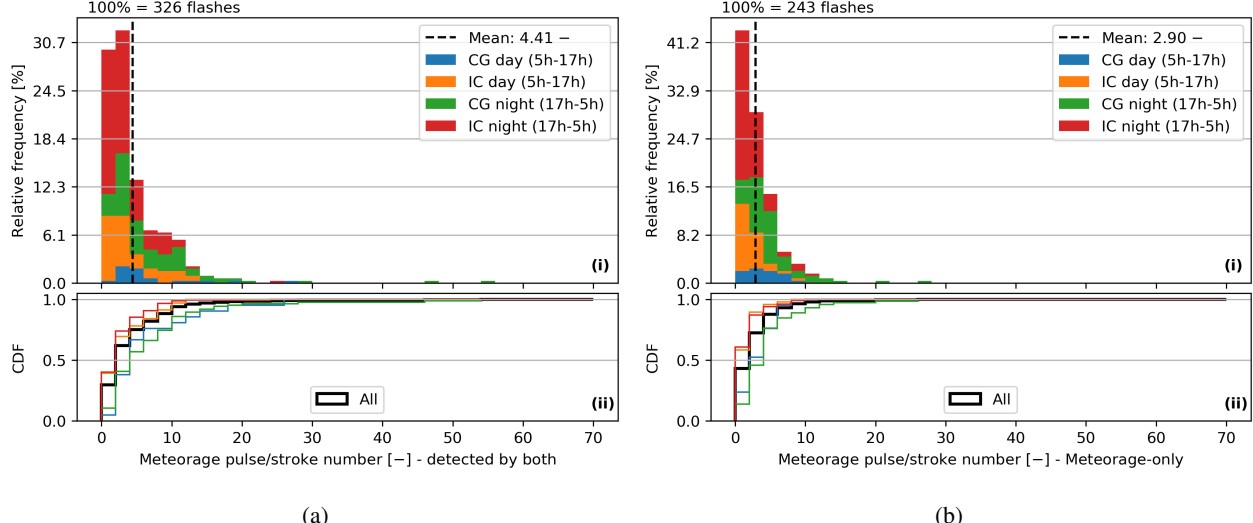

(a)                                                    (b)

**Figure 10.** Pulse/Stroke number of Meteorage flashes with coincident ISS-LIS flash (a) and unmatched Meteorage flashes (b). Daytime, nighttime and flash type, IC and CG, are indicated by the colors. Histogram (i) and corresponding CDF (ii) use the same colors. The CDF shows in addition a black curve for all data. The histogram bin width is constant at 2 pulses/strokes. The mean value is plotted as dashed line. The total number of flashes is indicate above the histogram.

in 82.7 % of the cases. It can be constituted that Meteorage better detected low altitude ISS-LIS flashes than ISS-LIS flashes restricted to mid and high levels.

The radiance of ISS-LIS flashes increases in general with the mean altitude (Figure 13(a)(i) and (b)(i)). The highest observed flash mean altitudes occur mainly for pure IC flashes and show statistically high radiances. They are likely within high reaching clouds like cumulus congestus and cumulonimbus. The average mean radiance for the matched and unmatched ISS-LIS flashes yields 18.2 $\mu J \cdot sr^{-1} m^{-2} \mu m^{-1}$ and 16.0 $\mu J \cdot sr^{-1} m^{-2} \mu m^{-1}$, respectively. Similar results regarding the mean radiance distributions can be identified for the ISS-LIS flash maximum altitude distributions (not shown).

Figure 14(i) illustrates the results of maximum radiance per flash within the maximum flash altitude histogram. The highest flashes do not necessarily contain the brightest events, however, the average trend is that maximum radiances slightly increase with the altitude of the flash. The maximum radiance distributions for the altitude bins show a wide spread. Medians of bins between 8 km and 13 km maximum flash altitude remain within the corresponding Inter-Quartile-Ranges (IQRs) of neighboring bins. The brightest event (127.0 $\mu J \cdot sr^{-1} m^{-2} \mu m^{-1}$) occurred during nighttime for a matched flash. The strongest
optical signal during the day (105.0 $\mu J \cdot sr^{-1} m^{-2} \mu m^{-1}$) is recorded within a matched flash, too. Accordingly, an average maximum radiance per flash of about 53.3 $\mu J \cdot sr^{-1} m^{-2} \mu m^{-1}$ and 36.9 $\mu J \cdot sr^{-1} m^{-2} \mu m^{-1}$ characterizes matched and unmatched flashes, respectively. Flashes containing the optically brightest events have a higher chance of producing significant LF signals and being detected by Meteorage than the optically dark flashes.





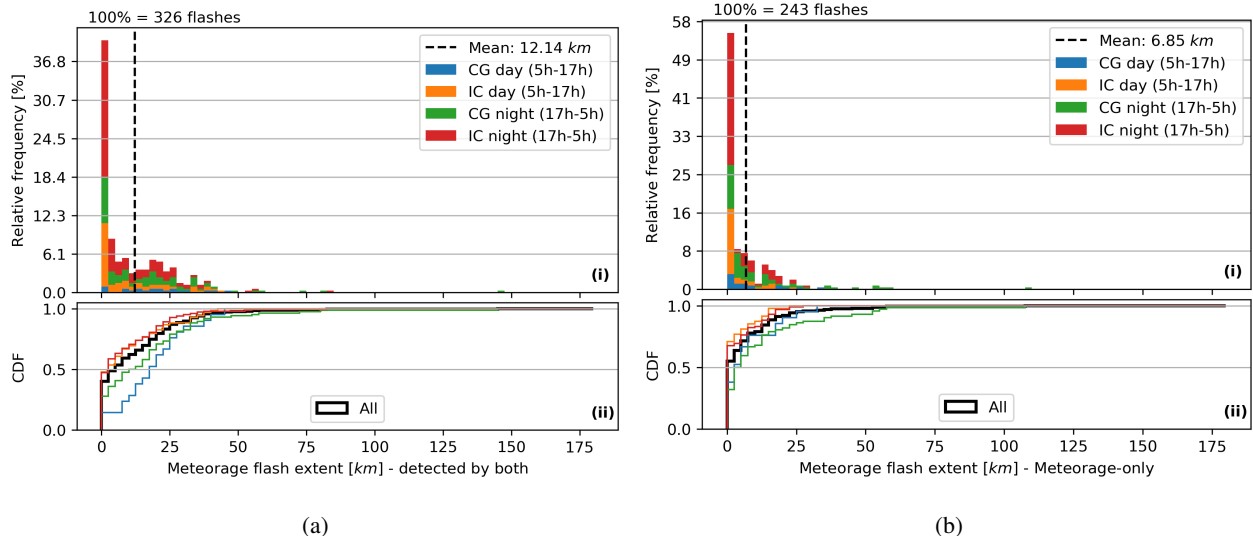

(a)           (b)

**Figure 11.** Flash extent of Meteorage flashes with coincident ISS-LIS flash (a) and unmatched Meteorage flashes (b). Daytime, nighttime and flash type, IC and CG, are indicated by the colors. Histogram (i) and corresponding CDF (ii) use the same colors. The CDF shows in addition a black curve for all data. The histogram bin width is constant at 2.5 km. The mean value is plotted as dashed line. The total number of flashes is indicate above the histogram.

The comparison of altitudes of Meteorage flashes with and without ISS-LIS matches aims at studying how ISS-LIS can detect
low altitude flashes. Flash mean (absolute) pulse/stroke amplitude and maximum amplitude per flash are additionally analyzed for each altitude bin in the histograms. The maximum amplitude per flash can either show a positive or negative current. Results are presented in Figure 15 for the altitude mean and in Figure 16 for the altitude maximum with the mean absolute pulse/stroke amplitude per flash. An average mean altitude of 8.1 km is found for Meteorage flashes with match (Figure 15(a)). It is about 1.4 km lower for the Meteorage-only flashes (Figure 15(b)). The mean altitude of matched flashes is similar to that
of ISS-LIS matched flashes (Figure 13(a)). The unmatched flashes, however, differ by about 1.5 km in altitude. Meteorage-only flashes occur at significantly lower altitude than the matched flashes(matched and unmatched ISS-LIS flashes are found at similar altitudes). Meteorage flash maximum altitudes confirm this result: Flashes with coincident ISS-LIS flash reach on average 9.8 km of altitude. The Meteorage-only flashes feature an average maximum altitude of 8.2 km. The maximum altitude distribution peaks, as for the ISS-LIS matched flashes, at about 11.0 km altitude (Figure 16). For the Meteorage-only flashes,
another mode exists between 6.5 km and 7.0 km of altitude. The low altitude mode is also found in the mean flash altitude distribution (at about 5.0 km in Figure 15). It is indicative of ISS-LIS' reduced DE of low altitude flashes. Meteorage flashes with maxima exceeding 10.0 km (248) are detected by ISS-LIS in 75.4 % of the cases. ISS-LIS' relative DE for Meteorage flashes with maxima lower (or equal) than 10.0 km (232) is only 45.3 %. This trend still influences the flash minima. Figure 17 shows the distribution of minimum flash altitudes with the maximum (pulse/stroke) amplitude per flash in each altitude bin.
Matched and unmatched flash average minimum altitudes approximate 6.1 km and 5.1 km, respectively. Hence, it is confirmed



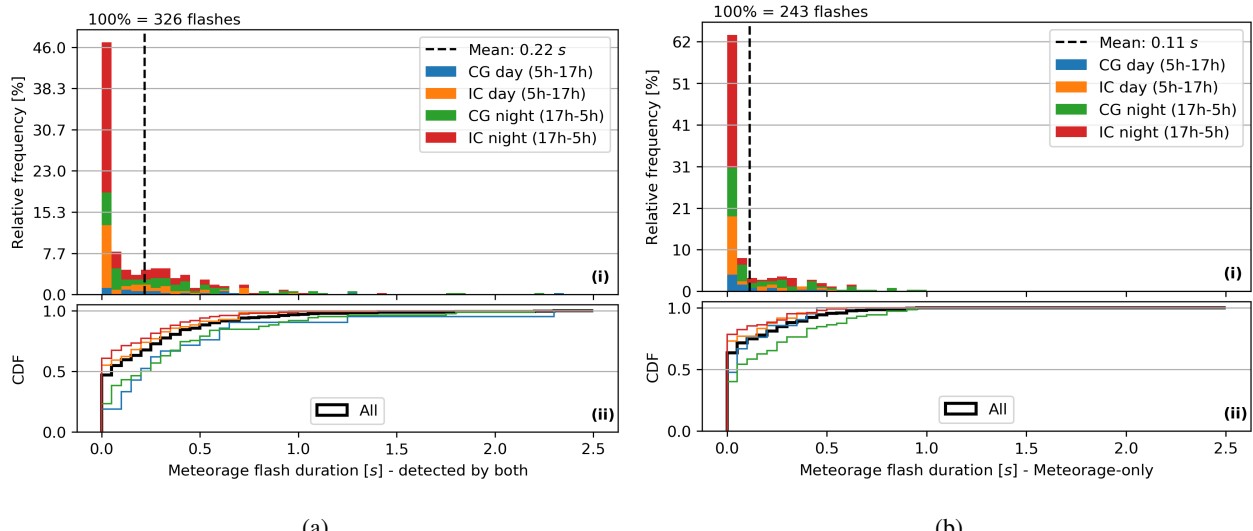

**Figure 12.** Meteorage flash duration with coincident ISS-LIS flash (a) and unmatched Meteorage flashes (b). Daytime, nighttime and flash type, IC and CG, are indicated by the colors. Histogram (i) and corresponding CDF (ii) use the same colors. The CDF shows in addition a black curve for all data. The histogram bin width is constant at 0.05 s. The mean value is plotted as dashed line. The total number of flashes is indicate above the histogram.

that ISS-LIS flash detection declines from high to low altitude flashes. The result agrees with the case study of (Thomas et al., 2000), who found significantly less skill of TRMM-LIS for (CG) discharges near the cloud base than for lightning channels propagating to near the top of the clouds.

Low altitude flashes (minimum altitudes below 5.0 km) feature statistically higher flash mean (not plotted) and maximum
amplitudes than flashes occurring above 5.0 km of altitude (Figure 17). Those flashes are mainly identified as CG flashes. The analysis of the flash maximum amplitude shows that those low altitude flashes are dominated by negative maximum currents. The flashes with minimum altitudes above 5.0 km exhibit statistically more positive than negative maximum currents. Further investigation reveals that about 94 % of the currents above 22.5 kA are observed for CG strokes. About 90 % of pulses/strokes with amplitude below 10.0 kA are IC pulses. CG strokes have almost exclusively negative currents in this study. Negative
currents are also observed for approximately 26 % of the IC pulses. The strongest currents reach up to 150.0 kA (both negative and positive currents) and occur for CG strokes. IC pulse currents do not exceed 50 kA.

The Meteorage mean (maximum) flash absolute amplitude equals 8.0 kA (13.2 kA) and 11.6 kA (18.1 kA) for matched and unmatched flashes, respectively. The difference between matched and unmatched flashes is attributed to some mid-level flashes producing strong currents and being not detected by ISS-LIS (difference between (a) and (b) in Figures 15 and 17).
However, the overall distributions of absolute flash amplitudes appear to be similar for matched and unmatched Meteorage flashes. One might use the polarity of the maximum current to identify whether the flash occurs below or above 5.0 km. This accounts for the flash minimum altitude. The relationship appears also for the flash maximum altitudes. Here, flashes with





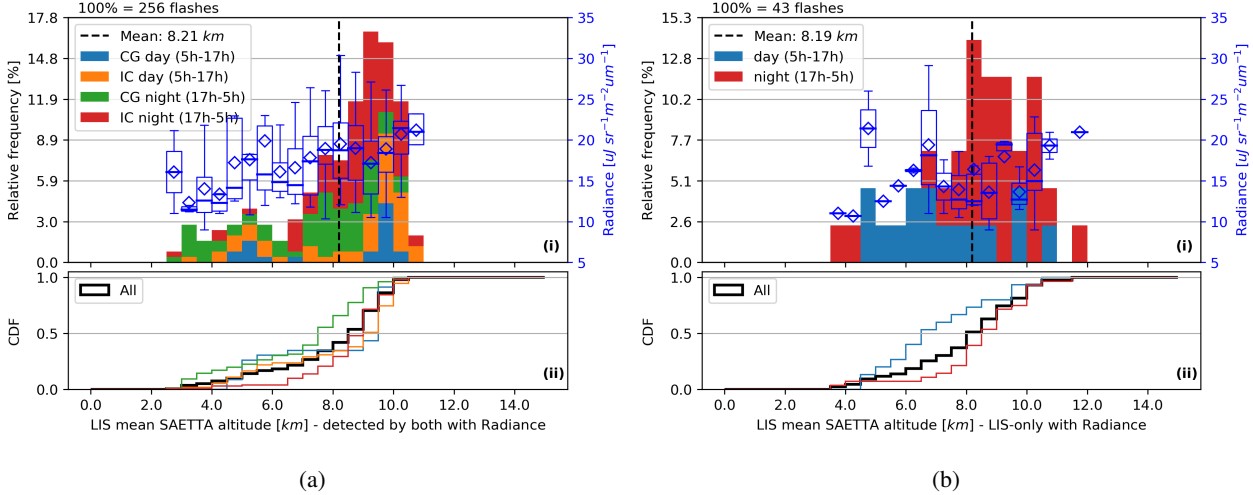

(a)                                         (b)

**Figure 13.** Flash mean altitude of ISS-LIS flashes (from concurrent SAETTA observations) with coincident Meteorage flash (a) and unmatched ISS-LIS flashes (b). Daytime, nighttime and flash type, IC and CG, are indicated by the colors. Histogram (i) and corresponding CDF (ii) use the same colors. The CDF shows in addition a black curve for all data. The histogram bin width equals 0.5 km. Note: The CG/IC attribute for ISS-LIS flash needs the matched Meteorage flash and does not exist for ISS-LIS-only flashes. The mean value is plotted as dashed line. The total number of flashes is indicate above the histogram. The blue boxplots (median as line, mean as diamond, Inter Quartile Range -IQR- as box, 1.5 IQR as whiskers - outliers not plotted) represent the distributions of ISS-LIS mean event radiance per flash for each altitude bin.

maximum altitudes below 10.0 km exhibit mainly negative maximum currents. It was investigated that ISS-LIS' DE is 30 % higher for flashes with maximum altitudes above 10.0 km than for flashes restricted to lower levels. Hence, the polarity of the

flash maximum current can provide a first information whether a flash is detected by ISS-LIS.

## 4    Conclusions

This study compares the results of the LF ground-based Meteorage LLS, the satellite sensor ISS-LIS and the VHF ground-based LMA SAETTA. The study domain is bounded to a region near Corsica in the Mediterranean Sea where SAETTA data are available. As ISS-LIS has been operating since March 2017, the period is confined to about one year from March 01, 2017

to March 20, 2018.

A new algorithm is developed to group both ISS-LIS events and Meteorage pulses/strokes to flashes. The algorithm is validated using concurrent SAETTA observations and the results of the existing NASA LIS algorithm.

ISS-LIS detects in total 16,881 events distributed over 330 flashes during its overpasses over the study domain. Meteorage data are filtered for the times of ISS overpasses. It contains 2,144 pulses/strokes (487 CG, 1657 IC) in 569 flashes. ISS-LIS

detects about 57.3 % of the Meteorage flashes. Especially CG-flashes and single pulse IC-flashes decrease the total relative



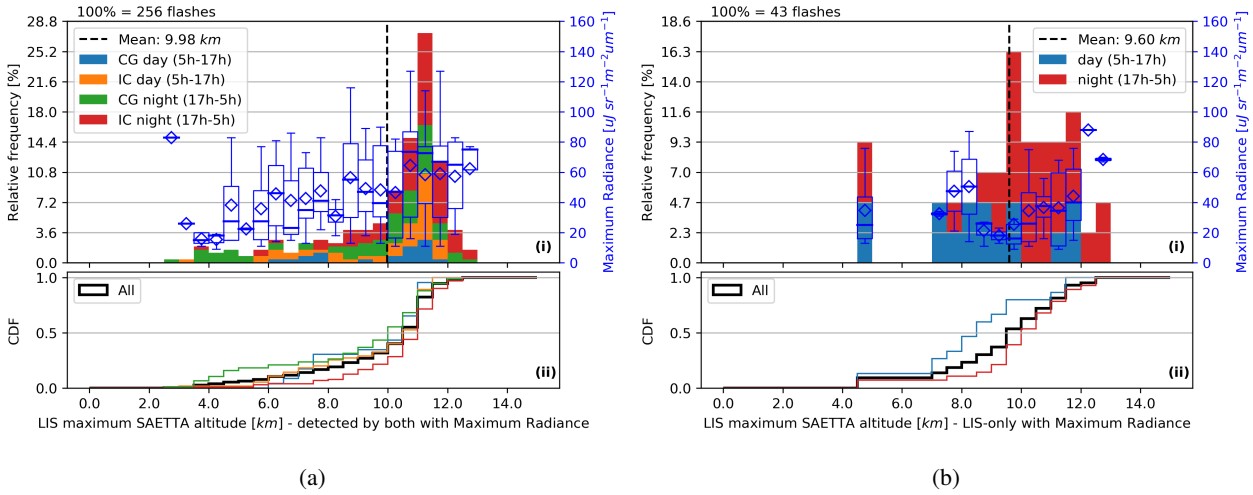

**Figure 14.** As Figure 13 for the maximum altitude of ISS-LIS flashes with (a) and without (b) match. Here, the blue boxplots (median as line, mean as diamond, IQR as box, 1.5 IQR as whiskers - outliers not plotted) represent the distributions of ISS-LIS maximum (event) radiance per flash for each altitude bin.

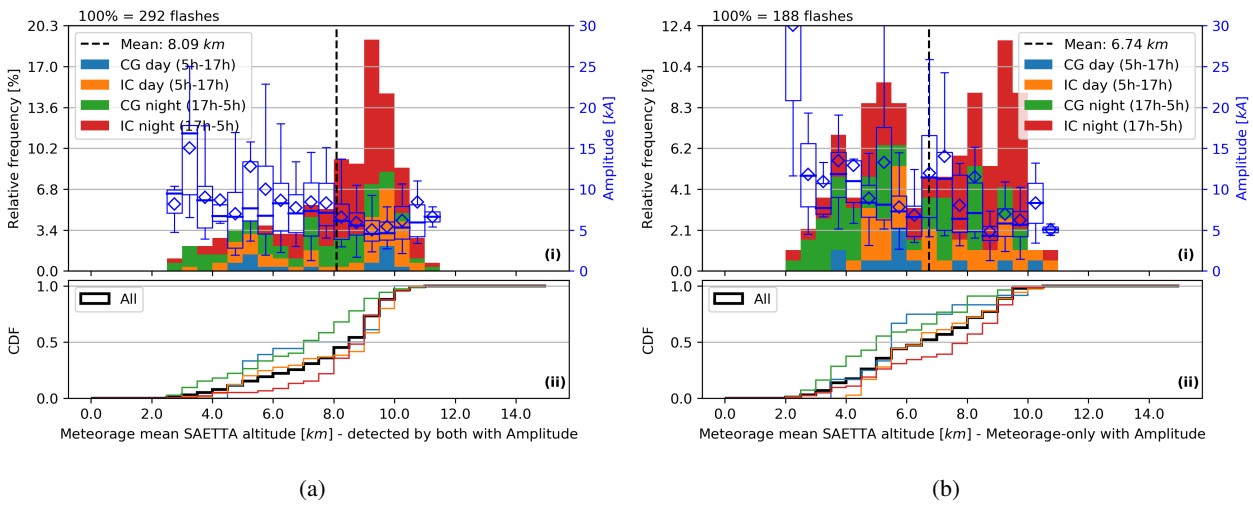

**Figure 15.** Flash mean altitude of Meteorage flashes (from concurrent SAETTA observations) with coincident ISS-LIS flash (a) and unmatched Meteorage flashes (b). Daytime, nighttime and flash type, IC and CG, are indicated by the colors. Histogram (i) and corresponding CDF (ii) use the same colors. The CDF shows in addition a black curve for all data. The histogram bin width equals 0.5 km. The mean value is plotted as dashed line. The total number of flashes is indicate above the histogram. The blue boxplots (median as line, mean as diamond, IQR as box, 1.5 IQR as whiskers - outliers not plotted) represent the distributions of Meteorage mean absolute pulse/stroke amplitude per flash for each altitude bin.



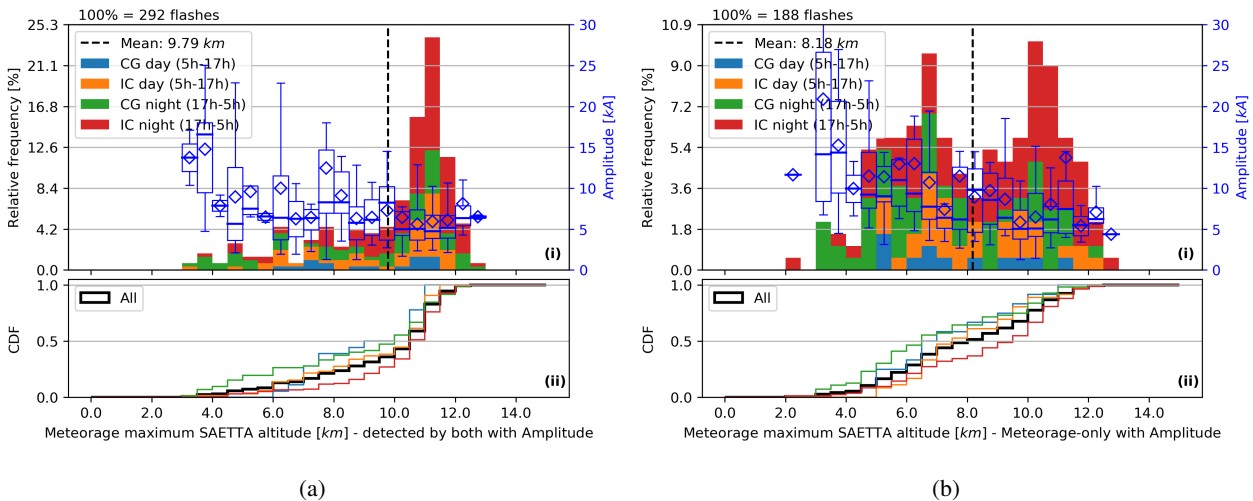

**Figure 16.** As Figure 11 for the maximum altitude of Meteorage flashes with (a) and without match (b).

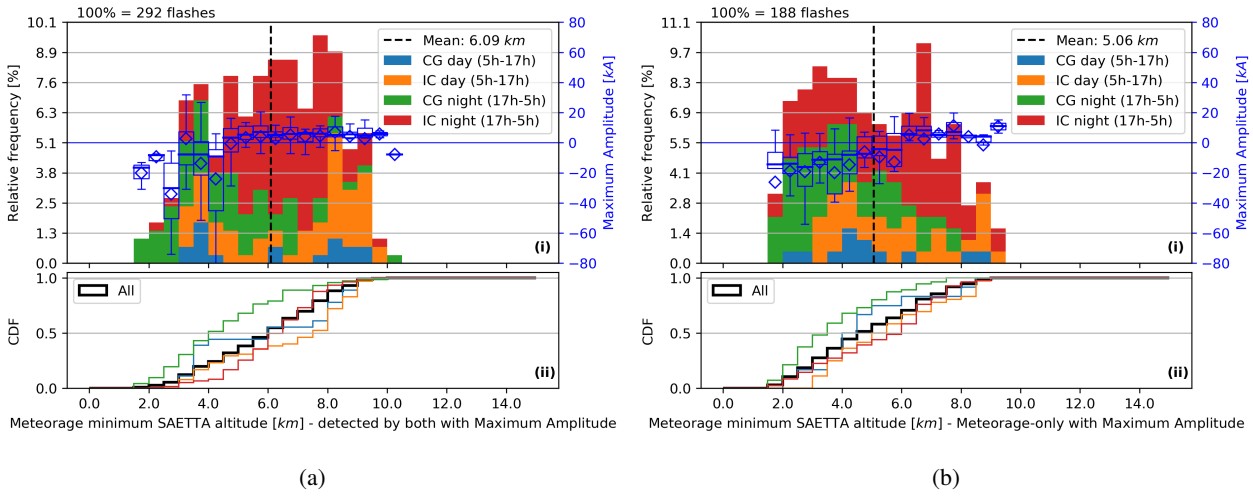

**Figure 17.** As Figure 11 for the minimum altitude of Meteorage flashes with (a) and without (b) match. Here, the blue boxplots (median as line, mean as diamond, IQR as box, 1.5 IQR as whiskers - outliers not plotted) represent the distributions of Meteorage maximum (pulse/stroke) amplitude per flash (positive or negative currents) for each altitude bin.

detection efficiency (DE) of ISS-LIS. A relative DE of 53.9 % is observed for flashes detected by Meteorage at daytime. LIS detected Meteorage intra-cloud/cloud-to-cloud (IC) flashes with about six percent higher relative DE than cloud-to-ground (CG) flashes. The LF Meteorage LLS is able to detect more than 80 % of all occurring ISS-LIS flashes.

Distances and timing offsets between matched ISS-LIS and Meteorage flashes are analyzed. A peak (median) distance (given
a Meteorage flash) of about 1.8 km (2.3 km) states a fairly accurate collocation of the flashes. Given an ISS-LIS flash, the peak





(median) distance equals about 3.0 km (4.7 km). It is generally smaller than the ISS-LIS pixel resolution (4.5 km nadir, 6.2 km at the edge of the field of view). The absolute timing offset distribution between a given Meteorage flash and the matched ISS-LIS flash is sharply peaked for less than 1.0 ms. Considering the ISS-LIS frame integration time of 2.0 ms, this is a very satisfying result. An analysis of the closest elements (events and pulses/strokes) reveals that, to an overall equal proportion,
615 ISS-LIS or Meteorage detect a lightning first while the peak timing offsets remain within the LIS frame integration time. For CG strokes, however, ISS-LIS tends to detect the lightning activity later than Meteorage. All offsets increase relatively from the distribution given a Meteorage flash to the distribution given an ISS-LIS flash. This finding is likely caused by the significantly lower number of pulses/strokes than the number of events. Thus, it is more likely to find an event close to a pulse/stroke than vice versa.

620 For an enhanced understanding of the flash detection by ISS-LIS and Meteorage, characteristics of the flashes are investigated. In accordance with e.g. Rudlosky et al. (2017), the probability of a match increases with larger flash extent and flash duration. A matched flash can extend on average almost twice as wide and last twice as long as a flash not seen by both ISS-LIS and Meteorage. In a similar manner, the matched flashes contain on average twice the number of elements than a flash observed by only one of the LLSs. ISS-LIS is sensitive to optical signals while Meteorage detects LF signals of electrical discharges.
625 Nevertheless, ISS-LIS flashes with at least one very bright event are more likely to be detected by Meteorage than optically dark flashes. Using the 3D lightning location of concurrent SAETTA observations, ISS-LIS and Meteorage flash altitudes are compared. Matched flashes of both ISS-LIS and Meteorage feature similar average mean altitudes near 8.2 km. Unmatched Meteorage flashes occurred on average 1.4 km lower than Meteorage flashes seen by ISS-LIS. Especially the maximum altitude of a flash influences significantly the detectability by ISS-LIS (compare e.g. Thomas et al. (2000)). Meteorage flashes with
630 maxima exceeding 10.0 km of altitude are detected by ISS-LIS in 75.4 % of the cases. ISS-LIS' relative DE for Meteorage flashes with maxima lower than 10.0 km of altitude is only 45.3 %. The Meteorage flash detection depends slightly on the flash minimum altitude. 89.7 % of the ISS-LIS flashes with minima less than 6.0 km of altitude have a coincident Meteorage flash. ISS-LIS flashes with minima above 6.0 km of altitude have a coincident Meteorage flash in 82.7 % of the cases.

 Further investigation revealed that the radiance of ISS-LIS flashes is somewhat correlated to the flash altitude with increasing
635 (both mean and maximum event) radiances for increasing flash altitudes. Meteorage amplitudes increase statistically with decreasing flash altitudes. Especially the polarity and the current of the strongest pulse/stroke within a Meteorage flash show potential to gain qualitative flash altitude information. Flashes with maximum currents of negative 10 kA or lower remain mainly below 10.0 km of altitude. As stated earlier, ISS-LIS' relative DE is 30 % higher for those flashes than for flashes with maximum altitudes above 10.0 km. This finding will need some additional proof, but it can be useful for mimicking satellite
640 lightning products using LF LLSs.

 This study analyzes satellite observed lightning over an extra-tropical region and compares the observations to ground-based LLSs. Our results including the statistics use about one year of data within the limited region around Corsica island. This results in a limited number of lightning cases. The limited region enables the direct unique comparison of not only ISS-LIS and LF Meteorage but also the VHF SAETTA LLS. Hence, ISS-LIS and Meteorage flash detection is investigated in more
645 detail, e.g. considering the concurrent SAETTA lightning source altitudes. The coincidences between ISS-LIS and Meteorage



flashes do not always have a one-to-one correspondence. It is, in addition, an artifact of the relatively coarse match constraints of 20.0 km in space and 1.0 s in time. The constraints are validated and their influence on the results is seen in the matched distance and timing offset distributions. It should be mentioned that the available ISS-LIS data is the provisional P0.2 version for this work. It is close to but not quite the fully validated data of ISS-LIS. Due to our limited number of cases, all ISS-LIS

data are treated in the same way independent of the position within ISS-LIS field of view (FOV). It is known that the ISS-LIS pixel (event) resolution and the DE decrease near the edge of the FOV. However, it is decided to not filter and reduce the observed cases further in order to allow a statistical analysis. Our method can be applied to geostationary satellite LLSs, i.e. GLM and the future MTG-LI, and the comparison of their observations to ground-based LLSs. It is planned to study GLM and NLDN lightning observations in America using our methodology. The geostationary satellite observes one region continuously

and thus there will be many more cases for the statistics. The results might be compared to our results of the comparison of ISS-LIS and Meteorage.

*Data availability.*   ISS-LIS provisional science data are available via NASA and HyDRO Search at the following doi:
https://doi.org/10.5067/LIS/ISSLIS/DATA204.
Fully validated ISS-LIS data are provided by NASA and HyDRO Search.

SAETTA data are available to members of HyMeX on the HyMeX website and can be provided on demand.
Meteorage data are provided by and property of Meteorage as company.

*Author contributions.*   FEr, EDe and OCa designed the methodologies to merge and analyze the lightning data. FEr implemented the methods and verified both code and results. FEr created all the plots. RJB, SPe and SCo provided the lightning data and some expertise on their quality. FEr wrote the manuscript. All authors contributed to revisions of the manuscript.

*Competing interests.*   The authors declare that they have no conflict of interest.

*Acknowledgements.*   FEr thanks the CNES and Météo-France for funding his PhD. This work is a contribution to the HyMeX programme through the EXAEDRE project, grant ANR-16-CE04-0005, funded by the French research foundation ANR and the SOLID project, funded by CNES. Acknowledgments are also addressed to CORSiCA-SAETTA main sponsors (Collectivité Territoriale de Corse through the Fonds Européen de Développement Régional of the European Operational Program 2007-2013 and the Contrat de Plan Etat Région,

HyMeX/MISTRALS, Observatoire Midi-Pyrénées, Laboratoire d'Aérologie, CNES) and many individuals and regional institutions in Corsica that host the 12 stations of the network or helped us to find sites. The authors also want to thank the SAETTA Team.





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
