# Peer review of "Concurrent Satellite and ground-based Lightning Observations from the Optical Lightning Imaging Sensor (ISS-LIS), the LF network Meteorage and the SAETTA LMA in the northwestern Mediterranean region"

_Atmospheric Measurement Techniques, 2019_

## Referee Comment (RC1) · Anonymous Referee #3 · 30 Jul 2019

General Comments

This manuscript compares the lightning detection characteristics of the satellite optical sensor (ISS-LIS) with a ground-based network of VLF/LF (electromagnetic) sensors (Meteorage) in preparation for future validation studies of the lightning imager to be onboard the MTG satellite. In order to better understand factors impacting the relative performance of these so-called LLS systems, the authors employed a ground-based

network of VHF sensors (SAETTA) that is capable of virtually 100% flash detection and 3-dimensional mapping of lightning channels within flashes. The authors chose to develop, evaluate, and employ their own algorithm for combining individual low-level reports provided by these LLS systems into lightning flashes, which are then matched in space and time to study the detection characteristics.

This work is a very thorough inter-comparison of these systems, along with a detailed assessment of performance relative to flash characteristics provided by the SAETTA system (flash duration, size, and height profiles). This work provides important information for the scientific community that is working to understand and employ satellite-based lightning observations. Overall, I view this manuscript as being well organized, and technically and scientifically sound. There are some very complicated methodologies and concepts in this work, so reading and understanding the content was sometimes hard work. This is made more difficult by the fact that the primary authors are not native English speakers. I also feel that some of the figures and associated discussions might not be necessary to support the important conclusions. It was difficult to understand the statistical analyses of factors impacting flash detection in Figures 13-17. As a last general comment, the labeling, size, and organization of many of the figure prevent them from readily supporting the findings described by the authors. None of these comments require specific changes by the authors – they are provided as personal opinions from a reviewer that is well-versed in all these systems and would like to see this work appreciated to its fullest potential.

I do have some specific comments, corrections, and recommendations that the authors need to address as they refine and revise this manuscript. This are followed be lesser editorial corrections and suggestions.

Specific Comments

1. Sentence on lines 47-48: networks with "widely-spaced" VLF/LF sensors (like Meteorage and the U.S. NLDN) report far more cloud pulses than return strokes, because

they are sensitive to vertically-oriented current-carrying channels in "larger" IC flashes (see Cummins and Murphy, 2009 or Nag et al., 2015). I would say that they have "…somewhat limited total lightning detection efficiency (DE)." Shorter-baseline VLF/LF systems like LINET and portions of the ENTLN (cited in this work) have very high total lightning flash DE.

2. Line 57: This is the first use of the term "relative DE", and (unfortunately) one of the referenced papers (Bitzer et al, 2016) use this term to mean the ratio of the conditional probabilities (see their equation (2)), but others do not use this definition. SO – you need to say that you use the common definition of relative DE used in many studies, which is the percentage of matched flashes divided by the number of flash in the other (reference) LLS (if that is what you are doing).

3. Line 138: the sentence about "…within 330 ms and 5.5 km…" is not really correct. No group associated with these values will be included in the flash – these are normalizing parameters for a Euclidian distance measure (see Mach et al., 2007 – figure 2)

4. Line 162: The Meteorage network only requires two sensors to report a lightning discharge (see Cummins and Murphy, 2009).

5. The description of LMA in lines 169-177 is not quite correct and should be reviewed by co-authors on your associated SAETTA paper. It reports leader development (associated with breakdown processes or fast leader propagation in established channels), serving to produce a spatial map of possible paths for later, high-current processes in the flash. The phrase "Fast CG discharges traveling between the cloud and ground" is not correct – you are probably referring to dart leaders that can occur in pre-established leader channels for earlier CG strokes, as well as in established channels within IC flashes.

6. Lines 211-213: regarding the authors' discussion of the rationale for using events rather than groups in the flash algorithm: This reviewer (and many others in the references cited in this work) firmly believes that reports from mid- to long-range ground-based VLF/LF LLS networks match well with LIS groups. Long vertical channels during periods of high currents provide localized light sources, and most of this light is produced during periods of less than 2 ms. The timing and centroid location of a LIS group are a good match for such sources. Figures 8 and 9 in Bitzer et al. (2016) show the very tight time- and space-correlations for these discharges. However, this does NOT mean that ALL LIS groups are space- and time-correlated with VLF/LF strokes/pulses. The authors' accompanying rationale related to lightning mapping and comparisons with SAETTA are quite reasonable. I ask that the authors refine/revise the rationale on these lines under the light of these comments.

7. Lines 284-292: regarding flash matching: This text seems to indicate that flashes are matched if any elements in the two LLS meet the time constraint, and if any elements (the same or different that the ones meeting the time constraint) meet the spatial constraint. This seems problematic, when multiple flashes are closely spaced in time and space. I ask that the authors clarify this point, to be sure that the algorithm is described properly.

8. Lines 397-403: The group:pulse distances should not necessarily be greater than the event:pulse distance, because the group locations are interpolated to sub-pixel spatial resolution by the radiance-weighting of the spatial centroid. Also the position differences reported by Bitzer et al. (2016) are much larger than the actual position differences because of +/- 5 km latitude location offsets associated with LIS yaw maneuvers (see Zhang et al., 2019). This issue is fixed in ISS-LIS. You may want to refine your analysis in the light of these issues.

9. Lines 469 and later: the units for the LIS radiance product is not correct, in terms of the spectral density. The units should be nm, not $\mu$m. See Zhang et al., 2019, which also shows that the TRMM-LIS minimum radiance is about 3 $\mu$JSr-1m-2nm-1, indicating that it has a lower threshold than ISS-LIS!

[Figure]

Editorial Corrections and Suggestions

10. Introduction, first paragraph: it might be helpful for some readers to also know that total lightning flash rate within a storm is associated with storm intensity features such as ice flux and updraft volume and rate (see Deierling and Petersen (2008) and Deierling et al., 2008, among others)

11. Sentence starting on line 41: "SAETTA" and "mapping" have not been introduced yet, and this sentence is probably not required at this point in the paper.

12. Line 50: the term "LIS groups" is used, but there has been no description of the LIS products (provided later in the manuscript). It might be helpful to do this in 2-3 sentences, or point the reader to section 2.1.

13. Line 90: suggest replacing "for GLM" to "as GLM"

14. Line 94: "orbits" should be "orbit", since there is only one orbit

15. Sentence starting on line 105 ("Among the..."), does not seen to be a complete sentence.

16. Line 112: suggest changing "... characteristics in lightning detection..." to "...lightning detection characteristics..."

17. Line 119: suggest changing "our paper" to "this work"

18. Lines 151-152: suggest changing "...from the lightning discharge on earth..." to be "...from the optical source at cloud-top..."

19. Line 196: The sentence starting on this line does not make sense to me. Two events in a flash can be much farther apart than 14.3 km. If you are referring to adjacent events then I do not see how they can be 11.9 km apart at nadir. Please clarify.

20. Line 204: suggest changing "group" to "collection", given the LIS definition of group

21. Line 222: Use of the words "initial element" suggests that all the sources must be

within the ds limit of the first source. This can cause spatially propagating flashes to be broken up. This should be clarified.

22. Line 299: change "onl yup" to "only up"

23. Section 3.2: Just a comment: It seems that the distance and timing offset distributions are produced by comparing all LIS events and all Meteorage pulse/strokes, so (for LIS) it mergers any underlying space:time correlations for individual matched pairs with the time-evolution of light that is observed by LIS, due to things like leader propagation and continuing current in long channels.

24. Lines 444-445: a contributing factor could also be related to increasing length of the optical sources due to finite leader and return-stroke velocities.

25. Lines 516-518: does this paragraph belong before the previous paragraph?

26. Line 528: The vertical displacement of LMA sources in the xlma display at large distances from the center of the network have always troubled me, even after speaking to the developer of xlma. It seems that refraction is not being handled, so that distant source heights are not really useable. Additional insights would be nice but are not required.

27. Line 543: suggest changing "It can be constituted that. . ." to "Overall, . . ."

28. The term "average mean radiance" (and similar terms that reflect statistics of statistics) require the reader to think hard to interpret the variables. Possible better wording could be "the mean radiance averaged over all heights" or something like that

29. Line 550: is it "maximum radiance per flash" or "maximum event radiance per flash" ?

30. Line 558: suggest changing "dark" to "darker", since they are not really dark (they can be seen).

31. Line 583: A reference for the 10 kA value would be helpful

32. 591-592: This seems to be an over-statement. The relationship between current, polarity, and height will vary with storm polarity and type, and falls apart for hybrid (IC+CG) flashes.

33. Conclusions section: Present tense should probably not be used, since the findings may not be universally applicable.

34. Line 614: it would improve clarity if you replaced "to an overall equal proportion" with "when considering the complete dataset"

35. Line 621: Zhang et al. (2019) might be a good reference to add here

36. Line 626: suggest replacing "dark" with "darker"

37. Lines 626:640: It might be useful to add that all of these height-related behaviors are likely driven by the range of heights associated with CG flashes vs. IC flashes.

References used in this review

Cummins, K., and M. Murphy (2009), An overview of lightning locating systems: History, techniques, and data uses, with an in-depth look at the US NLDN, IEEE Trans. Electromag. Compat., 51(3), 499–518, doi:10.1109/TEMC.2009.2023450

Deierling, W., W. A. Petersen, J. Latham, S. Ellis, and H. J. Christian (2008), The relationship between lightning activity and ice fluxes in thunderstorms, J. Geophys. Res., 113, D15210, doi:10.1029/2007JD009700

Deierling, W., and W. A. Petersen (2008), Total lightning activity as an indicator of updraft characteristics, J. Geophys. Res., 113, D16210, doi:10.1029/2007JD009598.

Mach, D., H. Christian, R. Blakeslee, D. Boccippio, S. Goodman, and W. Boeck (2007), Performance assessment of the Optical Transient Detector and Lightning Imaging Sensor, J. Geophys. Res., 112, D09210, doi:10.1029/2006JD007787.

Nag, A., M. J. Murphy, W. Schulz, and K. L. Cummins (2015), Lightning locating systems: Insights on characteristics and validation techniques, Earth and Space Science, 2, doi:10.1002/2014EA000051.

Zhang, D., K.L. Cummins, P. Bitzer, W.J. Koshak (2019), Evaluation of the Performance Characteristics of the Lightning Imaging Sensor, J. Atmos. Oceanic Tech., 36, 1015-1031, doi: 10.1175/JTECH-D-18-0173.

---

## Referee Comment (RC2) · Anonymous Referee #1 · 31 Jul 2019

**General Comments**

The authors provide a comprehensive overview of the relative performance of the Lightning Imaging Sensor and a traditional lightning location system (Meteorage), and use a research-grade VHF lightning mapping system (the SAETTA LMA) for reference. The authors apply a custom clustering methodology to all three datasets. The relative performance of these systems is of great interest beyond this specific project; the authors

note this fact and I agree with them.

Overall, the analysis method and results are clearly and comprehensively described, so this study only requires minor revisions. The study is relatively long, with dense prose summarizing each of the figures. The authors might consider trimming out some of the description of the figures in favor of (1) summarizing where things behave as expected, and (2) focusing on the most relevant or surprising findings.

Specific Comments

1. Check the order of introduction of instruments in Section 1 (Introduction) to make sure each instrument is described before being referred to by its acronym. There are also cross-references between instruments in odd places, such a mention of LF/VLF systems in the paragraph (line 80) discussing VHF systems. Overall, I thought the introduction could be shortened somewhat to focus more on the aims of the study, with less context about the lightning detection problem as a whole.

2. Line 84: WWLLN relies on ionospheric reflections but operates at VLF. As noted on line 164, the LF measurements in this study do not include ionospheric reflection.

3. Fig. 1c may be two lightning flashes in the VHF data, and if it was automatically identified by an algorithm illustrates the challenges in flash classification. There is a large gap in the channels to the SE, and if this were one flash I would expect to see that gap filled given the otherwise very well-resolved channels. Was there evidence of a new channel developing and exhibiting bidirectional development in the flash to the SE at ∼0.1 elapsed seconds?

4. Line 180: Recent studies by Chmielewski and Bruning (2016, 10.1002/2016jd025159) and Koshak, Mach and Bitzer (2018, 10.1175/JTECH-D-17-0041.1) show that changes to the network geometry can have a significant influence on detection efficiency and location precision. This effect may be important for the SAETTA network which has a long N-S baseline, in contrast to the somewhat

more circular and compact network in Thomas et al. (2004).

5. Lines 182-183: Does the study domain refer to the 350 km max range of detection or the somewhat smaller lat/lon box at the beginning of section 3?

6. Line 211: I don't have a concern here, so much as I wish to highlight that the authors raise an essential question about the measurements: "it is questionable whether LIS groups really correspond to (V)LF pulses/strokes." I agree with prior studies that the group is the fundamental physical measurable from the instrument - it is an ~instantaneous light emission tied to heating by a "large" current flow along a channel, and the events register the extent of the light scattered by that process. However, the authors are also right to point out that not all (V)LF pulses/strokes have a corresponding group, which suggests that either the instruments are sensitive to different physics, or the (V)LF and optical measurements lack the necessary sensitivity to see what is actually the same physics. In the end, the authors' approach of clustering using the events instead of the group centroids is a good choice, since considering events will help them align with LIS data and better identify coincident ground strokes that might happen at some distance from the centroid of the light emission as observed at cloud top, but I would disagree that the events are the fundamental physical detection.

7. In the paragraph beginning line 325, it is ambiguous whether the authors think the missed flashes nearest SAETTA were truly missed or if station downtime were to blame. This is especially interesting because the authors close the paragraph by stating that SAETTA is not an absolutely reliable DE reference. Pédeboy et al. (2018) is probably more explicit about the details, but it is not in a peer reviewed article; regardless it would be helpful to clarify here what the authors mean.

8. Given the predominance of flashes that occur outside SAETTA, altitude errors will be large and the total number of sources detected per flash will be small. How does this affect the results that depend on altitude retrieval from SAETTA in section 3.3?

9. Line 595: This statement is accurate for the authors' data, but I would predict storms

with inverted polarity would have the opposite expectation for detectability as function of polarity. This context would be helpful if another region were to be studied with the same methodology.

Technical Corrections

After revisions are completed I recommend an additional read for flow and a few missing words. For instance, on line 80: "uses very high frequency (VHF)" needs "radio signals" or some other noun at the end of the sentence.

The colon at the end of line 46 seems like it was from an earlier revision where the instruments were introduced in a different way?

Lines 609-13: "peak" here could be misinterpreted as "maximum." I suggest "mode" or "most frequent."

---

## Author Comment (AC1) · 25 Sep 2019

September 25, 2019

**1 AUTHORS RESPONSE TO EDITOR COMMENTS**

**RC1: Anonymous Referee #3, 30 Jul 2019**

The authors thank the referee #3 for the detailed and constructive comments. We included the general and specific comments in the updated paper manuscript. Based on the general comments, the results section is revised in total. Two Figures (former 14 and 15) are removed and we introduced two tables summarizing the statistics of investigated flash characteristics. The specific comments are addressed in the following.
**[We indicate the line number in the document showing track changes for each comment.]**

**1.1 Specific Comments**

1. Sentence on lines 47-48: networks with "widely-spaced" VLF/LF sensors (like Meteorage and the U.S. NLDN) report far more cloud pulses than return strokes, because they are sensitive to vertically-oriented current-carrying channels in "larger" IC flashes (see Cummins and Murphy, 2009 or Nag et al., 2015). I would say that they have "... somewhat limited total lightning detection efficiency (DE)." Shorter-baseline VLF/LF systems like LINET and portions of the ENTLN (cited in this work) have very high total lightning flash DE.

The statement is shortened and implemented after the introduction and citation of Cummins and Murphy (2009) and Nag et al. (2015).

Comment: The comment is true nowadays looking at the overall numbers of detected strokes/pulses (although it is different for the DE), however, our Meteorage data include no IC pulses until 2008. The first year exhibiting more IC pulses than CG strokes in Meteorage data was 2015 (within the study region of this work). Meteorage has still a higher (spatial) accuracy and DE for CG strokes than for IC pulses. The signal strengths (currents) are also significantly stronger for CG strokes than IC pulses. The predominant vertical channels within the cloud (Nag et al. 2015) do not exist for all IC discharges, thus their detection can be difficult for (V)LF LLSs. Return strokes associated with CG strokes do indeed very often produce sufficient (V)LF radiation to be measured by those LLSs (attenuation with distance to the discharge).

We want to point out that the detection efficieny (DE) of CG strokes is in general high for VLF/LF networks. The IC DE depends on the baseline distance. In particular, Meteorage is not a "widely-spaced" network as stated in the comment. The baseline distance is similar to that of e.g. LINET in Germany. The overall DE of Meteorage, with high CG DE and somewhat lower IC DE, is sufficient for a comparison to optical satellite instruments like ISS-LIS (it likely exceeds the ISS-LIS DE in most cases).

**[please see changes in lines 54-60]**

2. Line 57: This is the first use of the term "relative DE", and (unfortunately) one of the referenced papers (Bitzer et al, 2016) use this term to mean the ratio of the conditional probabilities (see their equation (2)), but others do not use this definition. SO – you need to say that you use the common definition of relative DE used in many studies, which is the percentage of matched flashes divided by the number of flash in the other (reference) LLS (if that is what you are doing).

The relative DE is briefly introduced prior to the literature review.
**[please see changes in lines 48-49]**

3. Line 138: the sentence about "...within 330 ms and 5.5 km..." is not really correct. No group associated with these values will be included in the flash - these are normalizing parameters for a Euclidian distance measure (see Mach et al., 2007 - figure 2)

The weighted Euclidean distance concept is introduced in the paper to correct the sentence. **[please see changes in lines 154-156]**

4. Line 162: The Meteorage network only requires two sensors to report a lightning discharge (see Cummins and Murphy, 2009).

We confirm that two sensors are sufficient. They use the time and angle of arrival to create four variables for three target values (time, latitude, longitude).
**[please see changes in line 182]**

5. The description of LMA in lines 169-177 is not quite correct and should be reviewed by co-authors on your associated SAETTA paper. It reports leader development (associated with breakdown processes or fast leader propagation in established channels), serving to produce a spatial map of possible paths for later, high-current processes in the flash. The phrase "Fast CG discharges traveling between the cloud and ground" is not correct – you are probably referring to dart leaders that can occur in pre-established leader channels for earlier CG strokes, as well as in established channels within IC flashes.

We removed the phrase "Fast CG discharges travelling between the cloud and ground in time frames shorter than 80 $\mu$s might be missed." It should indicate instead that flash components with continuous VHF radiation would be hardly mapped with a LMA due to time-of-arrival technique and the sampling method.
**[please see changes in line 194]**

6. Lines 211-213: regarding the authors' discussion of the rationale for using events rather than groups in the flash algorithm: This reviewer (and many others in the references cited in this work) firmly believes that reports from mid- to long-range ground-based VLF/LF LLS networks match well with LIS groups. Long vertical channels during periods of high currents provide localized light sources, and most of this light is produced during periods of less than 2 ms. The timing and centroid location of a LIS group are a good match for such sources. Figures 8 and 9 in Bitzer et al. (2016) show the very tight time- and space-correlations for these discharges. However, this does NOT mean that ALL LIS groups are space- and time-correlated with VLF/LF strokes/pulses. The authors' accompanying rationale related to lightning mapping and comparisons with SAETTA are quite reasonable. I ask that the authors refine/revise the rationale on these lines under the light of these comments.

The formulation was too stringent. The authors agree that LIS groups often represent similar physical phenomena (discharge processes) as (V)LF pulses/strokes. The significantly higher number of LIS groups than (V)LF sources shows that LIS groups can indicate additional discharge mechanisms that are not seen in the (V)LF range. The rationale is adapted.
**[please see changes in lines 243-244]**

7. Lines 284-292: regarding flash matching: This text seems to indicate that flashes are matched if any elements in the two LLS meet the time constraint, and if any elements (the same or different that the ones meeting the time constraint) meet the spatial constraint. This seems problematic, when multiple flashes are closely spaced in time and space. I ask that the authors clarify this point, to be sure that the algorithm is described properly.

The matching algorithm uses a combined distance-time criterion. The same elements must meet both the spatial and the time constraint. Flashes are only matched if at least two elements (one per flash) meet both the spatial and the temporal criteria for a match.
Section 2.5 is revised in total.
**[please see changes in lines 317-323]**

8. Lines 397-403: The group:pulse distances should not necessarily be greater than the event:pulse distance, because the group locations are interpolated to sub-pixel spatial resolution by the radiance-weighting of the spatial centroid. Also the position differences reported by Bitzer et al. (2016) are much larger than the actual position differences because of +/- 5 km latitude location offsets associated with LIS yaw maneuvers (see Zhang et al., 2019). This issue is fixed in ISS-LIS. You may want to refine your analysis in the light of these issues.

It is true that group centroids can be located in any location while event locations are fixed to the pixel centers. Group centroids can in fact be closer or further away from VLF/LF pulses/strokes (which are not bound to pixel centers obviously). One should expect statistically about the same distance between VLF/LF pulses/strokes and LIS events and groups. We removed the statement and added a short discription of the difference between event and group locations.

Zhang et al. (2019) provide very interesting findings and it is a valuable reference to be cited here. The correction during TRMM yaw maneuvers results in slightly lower distances to ground-based LLS for TRMM-LIS groups (1-2 km in Zhang et al.) than for ISS-LIS events in this study (2-4 km).
**[please see changes in lines 447-460]**

9. Lines 469 and later: the units for the LIS radiance product is not correct, in terms of the spectral density. The units should be nm, not $\mu$m. See Zhang et al., 2019, which also shows that the TRMM-LIS minimum radiance is about 3 $\mu$JSr-1m-2nm-1, indicating that it has a lower threshold than ISS-LIS!

The unit issue was further examined in collaboration with D. Buechler (University of Alabama Huntsville, Huntsville, AL, USA). We found that the available version of ISS-LIS P0.2 had not yet included the calibrated radiance. The radiance variable in the data has the same values as (and is therefore identical to) the uncalibrated (raw) optical amplitude count. Hence, all analyses of optical signal strength ("radiance") are actually amplitude counts. The paper terminology is changed and the corresponding Figure labels are updated.
**[please see changes in lines 162-166 and e.g. in lines 532-533]**

1.2 Editorial Corrections and Suggestions

10. Introduction, first paragraph: it might be helpful for some readers to also know that total lightning flash rate within a storm is associated with storm intensity features such as ice flux and updraft volume and rate (see Deierling and Petersen (2008) and Deierling et al., 2008, among others)

Suggested references were cited with a short summary of their work.
**[please see changes in lines 27-32]**

11. Sentence starting on line 41: "SAETTA" and "mapping" have not been introduced yet, and this sentence is probably not required at this point in the paper.

The second part of the sentence was indeed not required here and is removed.
**[please see changes in lines 46-47]**

12. Line 50: the term "LIS groups" is used, but there has been no description of the LIS products (provided later in the manuscript). It might be helpful to do this in 2-3 sentences, or point the reader to section 2.1.

Groups are briefly explained.
**[please see changes in line 64-65]**

13. Line 90: suggest replacing "for GLM" to "as GLM"

ok.
**[please see changes in line 105]**

14. Line 94: "orbits" should be "orbit", since there is only one orbit

ok.
**[please see changes in lines 109]**

15. Sentence starting on line 105 ("Among the...), does not seen to be a complete sentence.

The sentence was misleading and should be clear now.
**[please see changes in lines 120-122]**

16. Line 112: suggest changing "... chacteristics in lightning detection..." to "...lightning detection characteristics..."

ok.
**[please see changes in lines 127]**

17. Line 119: suggest changing "our paper" to "this work"

ok.
**[please see changes in lines 134]**

18. Lines 151-152: suggest changing "... from the lightning discharge on earth..." to be "... from the optical source at cloud-top..."

ok.
**[please see changes in lines 171-172]**

19. Line 196: The sentence starting on this line does not make sense to me. Two events in a flash can be much farther apart than 14.3 km. If you are referring to adjacent events then I do not see how they can be 11.9 km apart at nadir. Please clarify.

We tried to give a value for the distance constraint of the NASA ISS-LIS clustering algorithm to be compared to our in-house algorithm. We cannot really identify such a value between the events (additional communication to D. Mach [NASA MSFC, Huntsville, AL, USA]). The NASA ISS-LIS clustering algorithm uses a weighted Euclidian distance (space and time) between group centroids. The paper is corrected.
**[please see changes in lines 225-229]**

20. Line 204: suggest changing "group" to "collection", given the LIS definition of group

ok.
**[please see changes in line 235]**

21. Line 222: Use of the words "initial element" suggests that all the sources must be within the ds limit of the first source. This can cause spatially propagating flashes to be broken up. This should be clarified.

The term "the initial element" is replaced by "any element of the flash". The algorithm can indeed treat spatially propagating flashes.
**[please see changes in lines 255-259]**

22. Line 299: change "onl yup" to "only up"

ok.
**[please see changes in lines 342]**

23. Section 3.2: Just a comment: It seems that the distance and timing offset distributions are produced by comparing all LIS events and all Meteorage pulse/strokes, so (for LIS) it mergers any underlying space:time correlations for individual matched pairs with the timeevolution of light that is observed by LIS, due to things like leader propagation and continuing current in long channels.

We do not completely understand this comment. We would appreciate if you explain your comment further.

24. Lines 444-445: a contributing factor could also be related to increasing length of the optical sources due to finite leader and return-stroke velocities.

Do you mean we need to consider the time of propagation of the leader or return stroke to explain the time difference between the optical signal and the Meteorage CG stroke?

25. Lines 516-518: does this paragraph belong before the previous paragraph?

Yes, the brief summary fits better directly after the comparison and before discussing the exceptionally long flashes.
**[please see changes in lines 581-590]**

26. Line 528: The vertical displacement of LMA sources in the xlma display at large distances from the center of the network have always troubled me, even after speaking to the developer of xlma. It seems that refraction is not being handled, so that distant source heights are not really useable. Additional insights would be nice but are not required.

Currently the SAETTA data processing is similar to the other LMA processing chains and does not include any refraction correction. As a matter of fact electromagnetic waves propagating in the clear sky are downward deflected because of the refractive index gradient that is most often downward directed. If refraction is not taken into account in the calculation of the VHF source position, the calculated altitudes may overestimate the true altitudes for distant events from the network. This is something that should be investigated in the future.

27. Line 543: suggest changing "It can be constituted that..." to "Overall, ..."

Ok.
**[please see changes in line 618]**

28. The term "average mean radiance" (and similar terms that reflect statistics of statistics) require the reader to think hard to interpret the variables. Possible better wording could be "the mean radiance averaged over all heights" or something like that

Okay, the wording is clarified.
**[please see changes in e.g. lines 633-634]**

29. Line 550: is it "maximum radiance per flash" or "maximum event radiance per flash"?

It is the highest amplitude count (former stated as radiance) observed for an event of a flash. The term maximum event amplitude count per flash is more precise here. **[please see changes in e.g. lines 633-634]**

30. Line 558: suggest changing "dark" to "darker", since they are not really dark (they can be seen).

Ok.
**[please see changes in line 636]**

31. Line 583: A reference for the 10 kA value would be helpful

The value was inititally chosen to discuss the observed differences in the results. For example, Cummins and Murphy (2009) also use values of 10 kA and 20 kA to distinguish "small events" and "larger events", respectively, and to identify CG strokes and IC pulses.
**[please see changes in lines 675-681]**

32. 591-592: This seems to be an over-statement. The relationship between current, polarity, and height will vary with storm polarity and type, and falls apart for hybrid (IC+CG) flashes.

The possibility of a different relationship in inverted polarity storms (and in other regions) is added to the discussion in the paper. The relationship of the polarity of the maximum current and the flash maximum altitude might also be distorted for hybrid

flashes. The fact is added to the discussion in paper. We do not have data to prove this idea.
**[please see changes in lines 692-695]**

33. Conclusions section: Present tense should probably not be used, since the findings may not be universally applicable.

The use of present tense in the conclusions is revised and changed for study specific findings.
**please see changes in section 4, page 23**

34. Line 614: it would improve clarity if you replaced "to an overall equal proportion" with "when considering the complete dataset"

The author does not totally agree with the suggested wording. The study presents two distributions (for ISS-LIS and for Meteorage). The sentence provides the information that both distributions feature a similar ratio of negative values (indicating the match was detected prior to the source). The term "to an overall equal proportion" is replaced with "with similar probability".
**[please see changes in line 716]**

35. Line 621: Zhang et al. (2019) might be a good reference to add here

Ok.
**[please see changes in line 723]**
36. Line 626: suggest replacing "dark" with "darker"

Ok.
**[please see changes in line 728]**

37. Lines 626:640: It might be useful to add that all of these height-related behaviors are likely driven by the range of heights associated with CG flashes vs. IC flashes.

A short sentence is added mentioning the flash types.
**[please see changes in line 729-730]**

**2 ADDITIONAL TRACK CHANGES**

| Instance | Description |
|---|---|
| Figure 1 | Time series added (d) |
| Section 2.1 | Raw amplitude count introduced as optical signal measure |
| Section 2.5 | Revised and slightly shortened |
| Section 3.3 | Revised |
| Table 2 | New table added (section 3.3) |
| Table 3 | New table added (section 3.3) |
| Figure 13 | Labels updated (Radiance -> Amplitude count) |
| Figure 14 | Removed (information gain after Figure 13 is low) |
| Figure 15 | Removed (results in Table 3 and Figure 16) |

---

## Author Comment (AC2) · 25 Sep 2019

September 25, 2019

**1  AUTHORS RESPONSE TO EDITOR COMMENTS**

**RC2: Anonymous Referee #1, 31 Jul 2019**

The authors thank the referee #1 for his constructive comments. We included the general and specific comments in the updated paper manuscript. Based on the general comments, the results section is revised in total. Two Figures (former 14 and 15) are removed and we introduced two tables summarizing the statistics of investigated flash characteristics. The specific comments are addressed in the following.
**[We indicate the line number in the document showing track changes for each comment.]**

**1.1 Specific Comments**

1. Check the order of introduction of instruments in Section 1 (Introduction) to make sure each instrument is described before being referred to by its acronym. There are also cross-references between instruments in odd places, such a mention of LF/VLF systems in the paragraph (line 80) discussing VHF systems. Overall, I thought the introduction could be shortened somewhat to focus more on the aims of the study, with less context about the lightning detection problem as a whole.

The introduction was revised and refined. It should be clearer in terms of structure and the references. For example, papers of Cummins and Murphy (2009) and Nag et al. (2015) were added that provide summaries of the different LLSs. The introduction focuses more on the former studies and less on sensor characteristics than before. **[please see changes in section 1, pages 2-5]**

2. Line 84: WWLLN relies on ionospheric reflections but operates at VLF. As noted on line 164, the LF measurements in this study do not include ionospheric reflection.

This technical paragraph was shortened in order to focus on the literature review. **[please see changes in line 93-99]**

3. Fig. 1c may be two lightning flashes in the VHF data, and if it was automatically identified by an algorithm illustrates the challenges in flash classification. There is a large gap in the channels to the SE, and if this were one flash I would expect to see that gap filled given the otherwise very well-resolved channels. Was there evidence of a new channel developing and exhibiting bidirectional development in the flash to the SE at $\sim$0.1 elapsed seconds?

As a general comment the automatic algorithms to identify flashes from signals like VHF sources, LF pulses/strokes or optical events work in a statistical sense, however, neither the algorithms nor an observer can always distinguish individual flashes perfectly.

The flash in Figure 1c was identified as one flash by three different algorithms. Branches to both sides of the mentioned gap to the SE propagate towards each other. We looked at the flash development through a series of animations using SAETTA observations. We also looked in the raw LMA data to evaluate the idea given in the comment. In fact, there was a significant amount of VHF records and we could not clearly identify the single branches. Conclusively, we think that Figure 1c shows one flash that originated as two flashes whose channels connected. The connection is not directly visible in the data, as it is likely masked by the lack of reconstruction.
**[please see changes in Figure 1, page 28]**

4. Line 180: Recent studies by Chmielewski and Bruning (2016, 10.1002 / 2016jd025159) and Koshak, Mach and Bitzer (2018, 10.1175 / JTECH-D-17-0041.1) show that changes to the network geometry can have a significant influence on detection efficiency and location precision. This effect may be important for the SAETTA network which has a long N-S baseline, in contrast to the somewhat more circular and compact network in Thomas et al. (2004).

We agree with Referee #1 about the influence of the network geometry on the location precision. Coquillat et al. (2019) show that the displacement of 2 stations located in the south-west and south-east of Corsica towards 2 positions located in the extreme south and the extreme north of the island in 2016 led to a modification of the errors (Figure 3 in Coquillat et al., 2019). The radial error was markedly reduced except in the north of the studied domain meanwhile the altitude error increased almost everywhere at larger distances from the network. Therefore when different sets of at least 6 stations are involved in the calculation of the VHF source position one would expect a different geometry of the network, which influences the location precision. Nevertheless given the maximum SAETTA location errors (and errors within 50 km of the LMA) that are given in section 2.1, the location accuracy is sufficient for an intercomparison to both Meteorage and ISS-LIS as the maximum SAETTA location errors are not greater than the location errors of the LSSs in the comparison.

As far as detection efficiency is considered, Chmielewski and Bruning (2016) found in general high DE (>90 %) for all networks with different receiver thresholds up to a range of 150 km from the LMA center. Within a distance of 200 km, the modeled DE is still greater than 85 % in their study. The actual DE of SAETTA is difficult to address as the region and conditions are different from the regions used by Chmielewski and Bruning. One study is currently underway to assess SAETTA location accuracy through a comparison between SAETTA records and aircraft tracks in ice clouds.

The Koshak et al. (2018) paper is an interesting supplement and it is now given as a

reference in section 2.1.
**[please see changes in line 200-208]**

5. Lines 182-183: Does the study domain refer to the 350 km max range of detection
or the somewhat smaller lat/lon box at the beginning of section 3?

The study domain is limited to the region shown in Figure 1(a). It is introduced in the
opening paragraph of section 2 to be within 40.5 N to 44.0N and 7.0E to 11.0E. The
maximum distance to SAETTA's center approximates 271 km (the SW corner of the
domain) and 238 km (the NE corner of the domain).
The reference to Figure 1(a) was added to clarify the meaning of study domain.
**[please see changes in line 211]**

6. Line 211: I don't have a concern here, so much as I wish to highlight that the authors raise an essential question about the measurements: "it is questionable whether LIS groups really correspond to (V)LF pulses/strokes." I agree with prior studies that the group is the fundamental physical measurable from the instrument - it is an ~instantaneous light emission tied to heating by a "large" current flow along a channel, and the events register the extent of the light scattered by that process. However, the authors are also right to point out that not all (V)LF pulses/strokes have a corresponding group, which suggests that either the instruments are sensitive to different physics, or the (V)LF and optical measurements lack the necessary sensitivity to see what is actually the same physics. In the end, the authors' approach of clustering using the events instead of the group centroids is a good choice, since considering events will help them align with LIS data and better identify coincident ground strokes that might happen at some distance from the centroid of the light emission as observed at cloud top, but I would disagree that the events are the fundamental physical detection.

Yes, it is difficult to find a physical process representing the optical LIS events. Groups are likely the best representation of fundamental discharge processes in optical lightning data. Events should be seen as smallest, elementary measured units of LIS-like instruments. The rationale on these lines was refined (e.g., the word "fundamental" in line 234 was replaced by "elementary" to be consient with the further use of the term *element* for LIS events and LF pulses/strokes).
**[please see changes in lines 237, 243-244]**

7. In the paragraph beginning line 325, it is ambiguous whether the authors think the missed flashes nearest SAETTA were truly missed or if station downtime were to blame. This is especially interesting because the authors close the paragraph by stating that SAETTA is not an absolutely reliable DE reference. Pédeboy et al. (2018) is probably more explicit about the details, but it is not in a peer reviewed article; regardless it would be helpful to clarify here what the authors mean.

We looked in detail in our raw SAETTA data, with information from individual stations. It is found that VHF signals were observed for most of the missed (by SAETTA) flashes (114 of 120). The most important reasons for the classification as missed were signals at less than 6 stations and thus no reconstructed VHF source (77 of 120) and locating/timing differences just exceeding the algorithm criteria of 0.2 /0.3 s between observed flashes and VHF sources (34 of 120) or reconstructed VHF sources had too high reduced chi$^2$ values and were disregarded by our algorithm. As we do not know the origin of the signal in the raw data, we cannot assign raw data signals at individual stations to single flashes. This needs the reconstruction of VHF sources using at least 6 stations.

The missed flashes nearest SAETTA were not detected by SAETTA due to a reduced number of active stations (5 of 15), reconstructed SAETTA sources showed a slightly too high chi$^2$ uncertainty (5 of 15) or SAETTA reconstructed sources are found slightly shifted in space/time to a Meteorage flash (3 of 15). Only 2 Meteorage flashes in the proximity of SAETTA were really not detected.
**[please see changes in lines 368-383]**

8. Given the predominance of flashes that occur outside SAETTA, altitude errors will be large and the total number of sources detected per flash will be small. How does this affect the results that depend on altitude retrieval from SAETTA in section 3.3?

SAETTA detection range allows in general for many (order 100) sources per flash up to distances of about 350 km. The issue regarding the detection of low altitude sources in large distance to the LMA was added in the section 2.3. SAETTA is even capable of detecting sources lower than the theoretical values added in section 2.3, because most stations are well above sea level and the highest station is located at an altitude of 1950 mASL. In theory, this station has a direct line of sight to sources at about 1.6 kmASL altitude in a distance of 300 km to the station (which would be outside the study domain).

The altitude errors are given in section 2.3 and are expected to be less than 500 m within the entire domain (maximum distance to the LMA center of 270 km in the SW corner, significantly lower altitude errors closer to the network). The results in section 3.3 are discussed taking the theoretical altitude errors into account, however, actual errors also depend highly on the set of stations used for the reconstruction of the VHF source and the source altitude. This is especially true for each individual VHF source. The overall error of a sample of VHF sources should be more consistent and closer to the theoretical model than individual source errors. Flash altitudes in section 3.3 use statistical values of concurrent SAETTA sources (i.e. the mean, 10th and 90th percentile) and should represent the statistical flash altitudes within the given range of uncertainty.

[Figure]

9. Line 595: This statement is accurate for the authors' data, but I would predict storms with inverted polarity would have the opposite expectation for detectability as function of polarity. This context would be helpful if another region were to be studied with the same methodology.

It will need definitely further research to investigate how this relationship between polarity of the maximum (LF) current and the flash altitude behaves in different regions and different storm types. The suggested context is added to discussion in the paper. **[please see changes in line 692-695]**

1.2  Technical Corrections

After revisions are completed I recommend an additional read for flow and a few missing words. For instance, on line 80: "uses very high frequency (VHF)" needs "radio signals" or some other noun at the end of the sentence.

The entire paper is revised.

The colon at the end of line 46 seems like it was from an earlier revision where the instruments were introduced in a different way?

Yes, it was.
**[please see changes in line 53]**

X

Lines 609-13: "peak" here could be misinterpreted as "maximum." I suggest "mode" or "most frequent."

Ok.
**[please see changes in line 712]**

XI

**2  ADDITIONAL TRACK CHANGES**

| Instance | Description |
|---|---|
| Figure 1 | Time series added (d) |
| Section 2.1 | Raw amplitude count introduced as optical signal measure |
| Section 2.5 | Revised and slightly shortened |
| Section 3.3 | Revised |
| Table 2 | New table added (section 3.3) |
| Table 3 | New table added (section 3.3) |
| Figure 13 | Labels updated (Radiance -> Amplitude count) |
| Figure 14 | Removed (information gain after Figure 13 is low) |
| Figure 15 | Removed (results in Table 3 and Figure 16) |

**Supplement:**

[revised manuscript text omitted]

---

## Author Response (AR2)

**Paper Review**

Felix Erdmann

**AUTHORS RESPONSE TO EDITOR COMMENTS**

**1 Comments to Associate Editor Decision: Publish subject to minor revisions (review by editor) (07 Nov 2019) by Domenico Cimini**

The authors thank the reviewers and the associate editor for their careful reviews and suggestions.

General comment : We would like to mention that some line numbers of the last comments were not correct. We tried to identify the concerning phrases in the manuscript. Please see the track changes and indicate if changes were supposed to be done in different lines.

**1. The 2-sentence paragraph starting at line 47 does not flow well with the paragraphs before and after it. Maybe the authors could start this paragraph with something like "These comparisons focus on the spatial and temporal coincidence of flashes reported by the various systems, resulting in measures of detection efficiency (DE) as a function of various flash parameters. This study uses…"**

Suggestion accepted

**2. Line 84: suggest replacing "TRMM LIS records" with "the total number of observed flashes after combining the datasets."**

Suggestion accepted

**3. Line 101: suggest adding "VHF" before "Lightning Mapping"**

Suggestion accepted

**4. Line 291: suggest replacing "should also" with "can also"**

Suggestion accepted

**5. Line 320: It was not clear to me that this refinement was part of the algorithm until I read the whole paragraph. It might be helpful of the first sentence is modified to say "...offsets between matched flashes matching algorithm."**

We do not understand that comment and the suggested modification. The paragraph (line 304) starts with the phrase "The detailed analysis of distances and timing offsets between matched flashes refines the matching algorithm further". Nevertheless, after reviewing the text, we changed the order of the paragraphs starting in line 304 (about the refinement) and 314 (about the sensitivity study to determine ds_match and dt_match). The reading order is now similar to the algorithm workflow. The refinement happens after the general criteria are defined.

**6. Line 370: suggest replacing "missed flashes" with "flashes not reported by SAETTA"**

Suggestion accepted

**7. Line 499: "Figures" should not be capitalized**

It is changed.

**8. Line 501: should be LESS sensitive, not MORE sensitive**

Yes! It's indeed LESS sensitive during the day than during the night due to a higher threshold, and it is not the sensor itself but the acquisition of data.

**9. Line 502: text in parentheses is not clear to me**

1.0 is the observed difference of the minimum amplitude count during day and night, the overall minimum of 9.0 is given as a reference to value that difference. It is changed to give directly the threshold amplitude counts, 9.0 during the night and 10.0 during the day. We also give the number of nighttime events relying on that lower threshold (2,394 of 14,710).

**10. Line 511: extent looks more like 10 km in Figure 8**

5 km is actually a typo, it should be 15 km. We use a more precise value (13.2 km) to be consient in the paper. Table 1 shows the full list of calculated ISS-LIS flash extents.

**11. Line 518: extent looks more like 10 km, and only at night**

The difference exists at night and day (see the calculated values, Table 1). "About 5 km" was used as an estimation of the difference. It might be better to use a more precise value, 6.1 km.

**Table 1.** Flash extent for LIS flash types. All units are km.

| Type | Mean flash extent | Minimum flash extent | Maximum flash extent |
|---|---|---|---|
| LIS matched flashes | | | |
| CG day | 27.82 | 0.0 | 102.46 |
| IC day | 21.03 | 0.0 | 56.34 |
| CG night | 42.33 | 0.0 | 87.82 |
| IC night | 36.58 | 0.0 | 106.91 |
| LIS-only flashes | | | |
| day | 20.88 | 0.0 | 65.91 |
| night | 24.86 | 0.0 | 67.10 |

**12. Line 525: Does this second sentence indicate that ISS-LIS has higher DE that TRMM-LIS? Is this shown elsewhere in the paper? I do not think that this is correct**

The term "higher DE" does not refer to a comparison between ISS-LIS and TRMM-LIS. It rather refers to the different relative DE of ISS-LIS during day and night. The structure of sentence is sligthly changed.

**13. Table 3: the (signed) average maximum pulse/stroke amplitude does not seem to be a meaningful statistic. These three values COULD be replaced by "—" in the table**

Averaging over a distribution with positive and negative values is usually not very meaningful. In this case we kept the average value, as it is clearly negative and shows that negative and positive currents are not perfectly balanced (in Meteorage measurement observed here).

**14. Paragraph starting in Line 551: This content might be helpful earlier on the manuscript, when talking about Table 2 (or earlier – when discussion methods)**

The definition of flash mean, minimum and maximum amplitude can now be found at the end section 2.3.

**15. Line 77: suggest replacing "origin" with "originate"**

Suggestion accepted

III

**16. Line 609: the sentence starting near the end of this line ("CG strokes have . . .") does is not consistent with earlier content. If kept, it might be more helpful to sive specific counts.**

60   The sentence states "... almost exclusively ..." which is consistent with earlier statements. The vast majority of CG strokes had negative currents, however, a few strong, positive CG strokes occurred. Giving the actual percentage of negative CG strokes (90.6 %) is a good suggestion. It replaces the qualitative statement.

**ADDITIONAL TRACK CHANGES**

**Instance    Description**

-

[revised manuscript text omitted]

---

## Author Response (AR3)

**Paper Review**

Felix Erdmann

**AUTHORS RESPONSE TO EDITOR COMMENTS**

no new comments

**ADDITIONAL TRACK CHANGES**

[revised manuscript text omitted]